# T-JEPA: Augmentation-Free Self-Supervised Learning for Tabular Data

**Hugo Thimonier**[1,2][*]    **José Lucas De Melo Costa**[1][*]
**Fabrice Popineau**[1]    **Arpad Rimmel**[1]    **Bich-Liên Doan**[1]

[1] Université Paris-Saclay, CNRS, CentraleSupélec,
Laboratoire Interdisciplinaire des Sciences du Numérique,
91190, Gif-sur-Yvette, France.
[2] Emobot, France.
`{name}.{surname}@centralesupelec.fr`

## Abstract

Self-supervision is often used for pre-training to foster performance on a downstream task by constructing meaningful representations of samples. Self-supervised learning (SSL) generally involves generating different views of the same sample and thus requires data augmentations that are challenging to construct for tabular data. This constitutes one of the main challenges of self-supervision for structured data. In the present work, we propose a **novel augmentation-free SSL method** for tabular data. Our approach, T-JEPA, relies on a Joint Embedding Predictive Architecture (JEPA) and is akin to mask reconstruction in the latent space. It involves predicting the latent representation of one subset of features from the latent representation of a different subset within the same sample, thereby learning rich representations without augmentations. We use our method as a pre-training technique and train several deep classifiers on the obtained representation. Our experimental results demonstrate a substantial improvement in both classification and regression tasks, outperforming models trained directly on samples in their original data space. Moreover, T-JEPA enables some methods to consistently outperform or match the performance of traditional methods like Gradient Boosted Decision Trees. To understand why, we extensively characterize the obtained representations and show that T-JEPA effectively identifies relevant features for downstream tasks without access to the labels. Additionally, we introduce regularization tokens, a novel regularization method critical for training of JEPA-based models on structured data.

## 1 Introduction

Self-supervised learning has caught increasing attention in recent years due to its significant success in many applications. Self-supervision is often used for pre-training to improve models' performance on downstream tasks. In short, the objective of self-supervision for representation learning is to generate meaningful representations from unlabeled data by using pseudo-label. By pushing dissimilar samples farther away while reducing the distance between samples that are alike, self-supervised learning can facilitate learning for both supervised and unsupervised tasks.

Self-supervision often involves generating different *views* of the same sample to construct *positive* and possibly *negative* samples. The term positive sample designates samples related to one another, e.g., two pictures of a dog or the same picture cropped differently. In contrast, negative samples include unrelated samples, e.g., a picture of a cat and a dog. Given this terminology, two classes of self-supervised algorithms exist. Contrastive learning methods include negative and positive samples and non-contrastive learning techniques that rely exclusively on positive samples. Both approaches have offered promising results by generating meaningful representations of data (Chen et al., 2020; He et al., 2020; Tian et al., 2019; Assran et al., 2023) that allow the improvement of several models' performances on a broad range of tasks. Moreover, apart from improving supervised

---

[*]Equal contribution

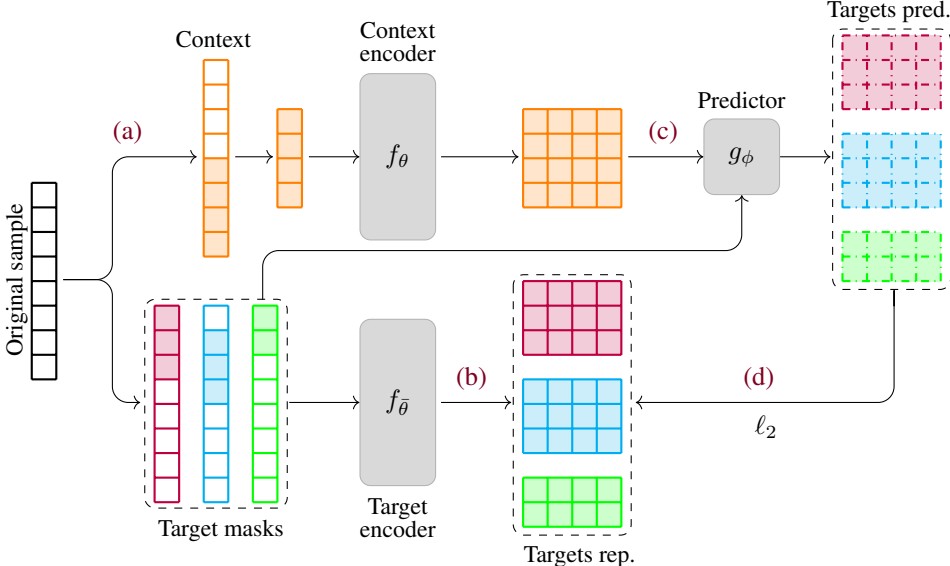

Figure 1: **T-JEPA training pipeline**. In step (a) a sample $\mathbf{x} \in \mathbb{R}^d$ is pre-processed and masked as detailed in equation 1 and fed to the context encoder to obtain a representation in $\mathbb{R}^{l_\mathbf{m} \times h}$ where $l_\mathbf{m}$ is the number of unmasked features for context mask $\mathbf{m}$. In step (b) the *unmasked* representation of sample $\mathbf{x}$ is fed to the target encoder and the features' representations are selected according to the corresponding target masks, as shown in equation 4 and Figure 7. In step (c) the output of the context encoder is fed to the predictor to obtain a prediction for each target mask used in step (b). In step (d) we compute the $\ell_2$-distance between the target representations and their predictions.

and unsupervised models' performance, (Hendrycks et al., 2019) have shown that self-supervision also helps improve robustness and uncertainty estimation for anomaly detection tasks.

Most self-supervised approaches include deep models, which have excelled in applications that include images or text. However, using neural networks for tabular data still remains challenging (Shwartz-Ziv and Armon, 2021). Indeed, Grinsztajn et al. (2022) discuss how the inherent heterogeneity of tabular data makes learning from this data structure using neural networks difficult. Nevertheless, recent work has investigated finding effective training procedures and novel architectures to learn from tabular data using neural networks. Recent advances include improved training procedures (Kadra et al., 2021; Gorishniy et al., 2021; Hollmann et al., 2023), representation learning for tabular data (Ye et al., 2024; Bahri et al., 2022; Zhu et al., 2023) or novel architectures (Somepalli et al., 2021; Kossen et al., 2021). Despite these recent advances, leveraging self-supervised learning for tabular data remains strenuous as most methods involve data augmentations to construct multiple views of the same data sample. While augmentations can be relatively straightforward for images or text data, constructing meaningful augmentation for tabular data is non-trivial. Augmentations for image samples often include cropping, rotation, or color alteration, while for text samples, this can include token masking or token replacement. These corruptions or augmentations of samples are domain-specific and hard to translate for structured tabular data as they can generate samples outside the data manifold.

As discussed by Assran et al. (2023), self-supervised learning for representation learning includes three types of approaches. First, Joint-Embedding Architecture (JEA) usually involves two encoders that learn to output similar embeddings for similar samples while ensuring distant embeddings for dissimilar samples. Second, Generative Architecture that aims at reconstructing a sample from a corrupted version of this sample, e.g., mask reconstruction. Third, Joint-Embedding Predictive Architecture (JEPA) resembles Generative Architecture as it relies on a similar task but in the latent space rather than the original data space. JEPA-based method consist in predicting a sample's representation in the embedding space from the embedded representation of a corrupted version of this sample. Recently, Assran et al. (2023) have proposed I-JEPA, a novel self-supervised approach targeted for images that does not involve augmentations. Their approach used for pre-training offered significant improvement in classification tasks on images. Following their path, other works have

extended their approach to video (Bardes et al., 2024), audio (Fei et al., 2024), and graphs (Skenderi et al., 2023). The present work aims to adapt JEPA for tabular data as a pre-training model to foster performance on classification and regression tasks. Recent work (Kossen et al., 2021; Ucar et al., 2021; Thimonier et al., 2024) has demonstrated that mask reconstruction can be a relevant pretext task for representation learning of tabular data. This work extends this mask reconstruction paradigm from the data space to the latent space. Adapting such approach to structured data is particularly relevant as it avoids constructing ad-hoc data augmentations that are challenging to construct for this data type.

The main contributions of our work are the following:

- We put forward **T**abular-**J**oint-**E**mbedding **P**redictive **A**rchitecture (T-JEPA), a novel **augmentation-free** self-supervised method for tabular data.

- T-JEPA offers significant **improvement in performance** for classification and regression tasks for tabular data. Moreover, we show that **augmented by T-JEPA some deep methods consistently outperform Gradient Boosted Decision Trees** on the tested datasets.

- We extensively characterize the obtained representations and provide explanation as to why our approach enhances performance on supervised tasks.

- We empirically uncover a **novel regularization method**, regularization tokens, that is critical to escape collapsed training regimes.

## 2 RELATED WORK

**Self-Supervised Learning for Representation Learning**    Representation learning consists in finding a transformation of the input data into a new feature space where relevant information is preserved or enhanced while noise and irrelevant details are filtered out or minimized. To that end, self-supervised approaches have become prevalent in the field. In the field of computer vision, methods like SwAV (Caron et al., 2020), VICReg (Bardes et al., 2022) or Barlow Twins (Zbontar et al., 2021) aim at producing two views of the same sample passed through two different networks, such that the outputs are maximally correlated. SwaV (Caron et al., 2020), for instance, aims at pushing the embeddings of different samples to belong to different clusters on the unit sphere. Barlow Twins (Zbontar et al., 2021) involve training two identical neural networks simultaneously on the same data but with different augmentations. The objective is to minimize the redundancy between the representations learned by each network while maximizing their agreement on the same input. VICReg (Bardes et al., 2022) encourages the model to focus on learning invariant features by explicitly modeling and minimizing the variance of feature embeddings. Other methods like MoCo (He et al., 2020) or SimCLR (Chen et al., 2020) focus on learning representations by contrasting positive and negative pairs. Other data structures have also benefited from representation learning using self-supervised approaches such as video (Jabri et al., 2020; Zhang and Crandall, 2022; Bardes et al., 2024), audio (Mittal et al., 2022; Niizumi et al., 2021; Korbar et al., 2018; Fei et al., 2024) or graph (Skenderi et al., 2023; You et al., 2020; Hwang et al., 2020).

Self-supervised methods can be categorized into three types of approaches: joint-embedding architectures, generative architectures, or joint-embedding predictive architectures. While the former two have been the most prevalent in the literature, recent work has demonstrated the potential of joint-embedding predictive architectures. Recently, I-JEPA (Assran et al., 2023) targeted for images has shown significant performance improvement over several self-supervised methods. Their approach was adapted to other data types such as video (Bardes et al., 2024), audio (Fei et al., 2024), and graphs (Skenderi et al., 2023) and proved to offer competitive performance in comparison with existing methods.

**Representation Learning for Tabular Data**    Representation learning for tabular data has caught increasing attention in recent years. Gorishniy et al. (2021) extensively investigate the benefits of pre-training models on tabular data to enhance performance. In other works, Somepalli et al. (2021) and Kossen et al. (2021) propose a pretraining procedure to foster the performance of their novel transformer-based architectures for tabular data. Parallel to that, some works have focused entirely on proposing self-supervised methods for representation learning of tabular data. One of the first approaches, VIME (Yoon et al., 2020), proposes to augment the existing reconstruction task with

estimating mask vectors from corrupted tabular data. Bahri et al. (2022) propose SCARF, a simple method based on contrastive learning in which different views of a sample are obtained by corrupting a random subset of features. Recent work has also investigated prototype-based representation learning for tabular data such as PTaRL (Ye et al., 2024). Other works, such as XTab (Zhu et al., 2023), TransTab (Wang and Sun, 2022) or UniTabE (Yang et al., 2024), propose self-supervised representation learning for cross-table pretraining. Wu et al. (2024) discuss the concepts of salient and mutual information and emphasize their key role in producing meaningful sample representations. They propose SwitchTab, which aims to foster the decoupling between the salient and mutual information contained in a sample to produce its representations. Lee et al. (2024) emphasize the necessity of correctly handling the heterogeneous features of tabular data. Somewhat close to our approach, their method consists in binning the values of each feature and proceeds to use as a pretext task the reconstruction of the bin indices rather than the value in the original feature space. Also related to our approach, Sui et al. (2024) propose an augmentation-free method that can be applied to tabular data. Their method consists in simultaneously reconstructing multiple randomly generated data projection functions to generate a data representation. Finally, most related to our method, Ucar et al. (2021) propose SubTab that divides the input features to multiple subsets to perform a pretext task close to mask reconstruction. The core difference with T-JEPA lies in the fact SubTab performs mask reconstruction in the original dataspace while T-JEPA performs this task in the embedded space. We provide in Appendix F.2 a more comprehensive description of each SSL methods relevant to the present work.

## 3 METHOD

As displayed in Figure 1, T-JEPA involves three main modules used to learn the final representation: a context encoder, a target encoder, and a prediction module. In short, we predict from a subset of features of a sample $\mathbf{x}$ the latent representation of another non-overlapping subset of features of $\mathbf{x}$. The context encoder is used for the prediction, while the target encoder is used to construct the representations to be predicted.

Formally, let $\mathbf{x} \in \mathcal{X} \subseteq \mathbb{R}^d$ be a sample with $d$ features, which can be either numerical or categorical. Let $h$ designate the hidden dimension of the transformer encoders, $f_\theta$ the context encoder, $f_{\bar{\theta}}$ the target encoder and $g_\phi$ the predictor.

**Embedding Layers and Masking**  Before being fed to the different modules, data is pre-processed using embedding layers. We normalize numerical features to obtain $0$ mean and unit variance, while we use one-hot encoding for categorical features. At this point, each feature $j$ for $j \in \{1, ..., d\}$ has an $e_j$-dimensional representation, $\mathbf{E}(\mathbf{x}_j) \in \mathbb{R}^{e_j}$, where $e_j = 1$ for numerical features and for categorical features $e_j$ corresponds to their cardinality. Each sample is accompanied by a masking vector $\mathbf{m} \in \{0, 1\}^d$ in which each entry designates whether a feature is masked: $\mathbf{m}^j = \mathbb{1}\{\text{feature } j \text{ is masked}\}$, where $\mathbb{1}\{\cdot\}$ is the indicator function. When masked, we drop the corresponding feature, and only keep the remaining unmasked features. For a mask $\mathbf{m}$ with $l_\mathbf{m}$ unmasked features, i.e. $d - \|\mathbf{m}\|_1 = l_\mathbf{m}$, sample $\mathbf{x}$ has the following embedded representation

$$\tilde{\mathbf{E}}(\mathbf{x}) = \{\mathbf{E}(\mathbf{x}_j) : i \in \{1, ..., d\}, \mathbf{m}^j = 0\}. \tag{1}$$

Each of the $d$ features is equipped with a learned linear layer, $\texttt{Linear}(e_j, h), \forall j \in \{1, \ldots, d\}$, that embeds the $e_j$-dimensional representation into an $h$-dimensional space. We pass each of the $l_\mathbf{m}$ unmasked features' encoded representations through their corresponding linear layers. We also learn $h$-dimensional index and feature-type embeddings following standard practice when leveraging transformers for tabular data. Both are added to the embedded representation of sample $\mathbf{x}$. Let $\mathbf{z}_\mathbf{x}^\mathbf{m} \in \mathbb{R}^{l_\mathbf{m} \times h}$ denote the obtained embedded representation of sample $\mathbf{x}$ with mask $\mathbf{m}$.

**Regularizing Token**  We also include a regularizing token [REG] inspired from the register token first proposed in (Darcet et al., 2024) for ViT's. We append this token to the obtained $\tilde{\mathbf{E}}(\mathbf{x})$ representation displayed in equation 1. This token is also equipped with a learned embedding layer. This token is only used to train T-JEPA and is discarded when training supervised classifiers on the downstream task. See Fig 7 in Appendix H for an illustration. We later discuss the necessity of including such token in section 5.2 and observe that it acts as a regularizing method to escape training regimes leading to representation collapse. For simplicity, we do not explicitly include the

regularizing token in the rest of the method description hereafter. See Appendix H for a more detailed discussion on regularization tokens.

**Masking strategy** The masking strategy differs between the context and target encoders. We sample several masks for each sample. Context masks are used to mask the samples *before* feeding them to the embedding module and context encoder. On the contrary, the target masks are used to construct the target representation *after* passing them through the embedding module and target encoder. Note that in both context and target masking, the regularizing token is never masked. Hence, the input of the context encoder is a masked representation of a sample $\mathbf{x}$, $\mathbf{z_x^m}$. In contrast, the target encoder receives as an input the embedded representation $\mathbf{z_x^{0_d}}$, where $\mathbf{0}_d$ is the $d$-dimensional null vector. Then, the target mask is used to mask the corresponding encoded feature representations of $\mathbf{z_x^{0_d}}$ as shown in equation 4. For both context and target encoders, we set a minimum and maximum share of features to be masked simultaneously and randomly sample a share in that interval. Let $M_{\text{context}}, M_{\text{target}}$ denote the set of sampled context and target masks, respectively. We construct $M_{\text{context}}, M_{\text{target}}$ such that intra-overlaps are permitted (masks from the same set can overlap), but inter-overlaps are not permitted (a mask from $M_{\text{context}}$ cannot overlap with a mask from $M_{\text{target}}$).

**Context and Target Encoders** The context encoder $f_\theta$ is a transformer encoder composed of $\ell$ layers and $k$ attentions heads. The context encoder relies on multi-head self-attention to produce meaningful representations for each sample. It receives as input a mask representation $\mathbf{z_x^m} \in \mathbb{R}^{l_\mathbf{m} \times h}$ and outputs a representation of similar dimension. The target encoder's architecture exactly reproduces the one of the context encoder. Let $f_{\bar\theta}$ denote the target encoder which receives as input $\mathbf{z_x^{0_d}}$ an unmasked embedded representation of sample $\mathbf{x}$. Like the context encoder, it outputs a representation of the same dimension as its input.

$$h_{\text{context}}^\mathbf{m} = f_\theta(\mathbf{z_x^m}) \in \mathbb{R}^{l_\mathbf{m} \times h} \quad \text{(context)} \tag{2}$$

$$h_{\text{target}} = f_{\bar\theta}(\mathbf{z_x^{0_d}}) \in \mathbb{R}^{d \times h} \quad \text{(target)} \tag{3}$$

Let $h_{\text{target}}^{\mathbf{m}_k}$ denote the masked target representation for mask $\mathbf{m}_k$, obtained by discarding the masked features' representations from $h_{\text{target}}$ as done in equation 1,

$$h_{\text{target}}^{\mathbf{m}_k} = \{h_{\text{target}}^{(i)} : i \in \{1, ..., d\}, \mathbf{m}_k^i = 0\}, \tag{4}$$

where $h_{\text{target}}^{(i)}$ is the $h$-dimensional representation of the $i$-th feature in the target vector $h_{\text{target}}$. Following previous work (Assran et al., 2023), The parameters of the context encoder, $\theta$, are learned through gradient-based optimization. In contrast, the parameters of the target encoder $\bar\theta$ are updated via an exponential moving average (EMA) of the context encoder's parameters.

**Predictor** The predictor $g_\phi$ is also set to be a transformer encoder whose weights are conjointly learned with the context encoder's weights through gradient-based optimization. The predictor's hidden dimension is downsized from the encoders' dimension $h$, using a linear layer. The predictor takes as input $h_{\text{context}}^{\mathbf{m_j}}$, the output of the context encoder for mask $\mathbf{m}_j \in M_{\text{context}}$, and a target mask $\mathbf{m}_k \in M_{\text{target}}$ designating which features to be predicted, $g_\phi(h_{\text{context}}^{\mathbf{m_j}}, \mathbf{m}_k)$. We parameterize the mask tokens in $\mathbf{m}_k$ by a learnable vector to which is added a positional embedding. Each context output is passed $|M_{\text{target}}|$ times through the predictor module to predict the corresponding feature representation for each target mask.

**Loss** The loss function used to optimize the weights $\theta, \phi$, of the context encoder and predictor respectively, is the $\ell_2$-distance between the reconstructed representation, $g_\phi(h_{\text{context}}^\mathbf{m}, \mathbf{m}_k)$ and the target representation $h_{\text{target}}^{\mathbf{m}_k}$,

$$\mathcal{L}(\mathbf{x}; M_{\text{context}}, M_{\text{target}}) = \frac{1}{|M_{\text{target}}|} \cdot \frac{1}{|M_{\text{context}}|} \sum_{\mathbf{m} \in M_{\text{context}}} \sum_{\mathbf{m}_k \in M_{\text{target}}} \left\| g_\phi(h_{\text{context}}^\mathbf{m}, \mathbf{m}_k) - h_{\text{target}}^{\mathbf{m}_k} \right\|_2^2 . \tag{5}$$

## 4 EXPERIMENTS

### 4.1 EXPERIMENTAL SETTING

**Datasets** Following previous work (Ye et al., 2024), we experiments on 7 datasets with heterogeneous features to test the effectiveness of T-JEPA. We test our approach on several supervised

Table 1: **Performance metrics** for different downstream models trained on the original dataspace and the generated T-JEPA representation across datasets. We also include for completeness the performance of XGBoost (Chen and Guestrin, 2016) and CatBoost (Prokhorenkova et al., 2018) as a baseline. We report an average over 20 runs and the corresponding standard deviation. We report in **bold** the metric that wins between the raw data representation and the augmented representations. We underline the overall best metric for a dataset.

| | AD ↑ | HE ↑ | JA ↑ | AL ↑ | CA ↓ | HI ↑ | MNIST ↑ |
|---|---|---|---|---|---|---|---|
| **Baseline Neural Networks** | | | | | | | |
| MLP | $0.827$ $\pm 1e^{-3}$ | $0.353$ $\pm 1e^{-3}$ | $0.672$ $\pm 1e^{-3}$ | $0.916$ $\pm 4e^{-3}$ | $0.511$ $\pm 3e^{-3}$ | $\mathbf{0.681}$ $\pm 4e^{-3}$ | $0.978$ $\pm 2e^{-3}$ |
| +T-JEPA | $\mathbf{0.866}$ $\pm 2e^{-3}$ | $\mathbf{0.400}$ $\pm 4e^{-3}$ | $\underline{\mathbf{0.728}}$ $\pm 3e^{-3}$ | $\mathbf{0.961}$ $\pm 6e^{-3}$ | $\mathbf{0.468}$ $\pm 4e^{-2}$ | $0.517$ $\pm 8e^{-2}$ | $\underline{\mathbf{0.983}}$ $\pm 1e^{-3}$ |
| DCNv2 | $0.829$ $\pm 4e^{-3}$ | $0.347$ $\pm 3e^{-3}$ | $0.662$ $\pm 3e^{-3}$ | $0.905$ $\pm 3e^{-3}$ | $0.504$ $\pm 4e^{-3}$ | $\mathbf{0.683}$ $\pm 3e^{-3}$ | $0.971$ $\pm 1e^{-3}$ |
| +T-JEPA | $\mathbf{0.861}$ $\pm 2e^{-3}$ | $\mathbf{0.399}$ $\pm 3e^{-3}$ | $\mathbf{0.723}$ $\pm 2e^{-3}$ | $\mathbf{0.955}$ $\pm 2e^{-3}$ | $\underline{\mathbf{0.420}}$ $\pm 3e^{-2}$ | $0.525$ $\pm 8e^{-2}$ | $\mathbf{0.981}$ $\pm 2e^{-3}$ |
| ResNet | $0.814$ $\pm 7e^{-3}$ | $0.351$ $\pm 2e^{-3}$ | $0.666$ $\pm 3e^{-3}$ | $0.919$ $\pm 2e^{-3}$ | $0.534$ $\pm 2e^{-3}$ | $0.674$ $\pm 4e^{-3}$ | $0.979$ $\pm 1e^{-3}$ |
| +T-JEPA | $\mathbf{0.865}$ $\pm 3e^{-3}$ | $\underline{\mathbf{0.401}}$ $\pm 2e^{-3}$ | $\mathbf{0.718}$ $\pm 3e^{-3}$ | $\underline{\mathbf{0.964}}$ $\pm 1e^{-3}$ | $\mathbf{0.441}$ $\pm 8e^{-2}$ | $\mathbf{0.705}$ $\pm 5e^{-3}$ | $\underline{\mathbf{0.983}}$ $\pm 2e^{-3}$ |
| AutoInt | $0.823$ $\pm 1e^{-3}$ | $0.338$ $\pm 3e^{-3}$ | $0.653$ $\pm 6e^{-3}$ | $0.894$ $\pm 2e^{-3}$ | $0.501$ $\pm 3e^{-3}$ | $\mathbf{0.694}$ $\pm 3e^{-3}$ | $0.901$ $\pm 6e^{-3}$ |
| +T-JEPA | $\mathbf{0.866}$ $\pm 2e^{-3}$ | $\mathbf{0.351}$ $\pm 3e^{-3}$ | $\mathbf{0.710}$ $\pm 3e^{-3}$ | $\mathbf{0.938}$ $\pm 4e^{-3}$ | $\mathbf{0.448}$ $\pm 2e^{-2}$ | $0.517$ $\pm 8e^{-2}$ | $\mathbf{0.978}$ $\pm 1e^{-3}$ |
| FT-Trans | $0.821$ $\pm 7e^{-3}$ | $0.363$ $\pm 2e^{-3}$ | $0.677$ $\pm 2e^{-3}$ | $0.913$ $\pm 3e^{-3}$ | $0.473$ $\pm 5e^{-3}$ | $\mathbf{0.684}$ $\pm 5e^{-3}$ | $0.811$ $\pm 5e^{-2}$ |
| +T-JEPA | $\mathbf{0.864}$ $\pm 1e^{-3}$ | $\mathbf{0.384}$ $\pm 5e^{-3}$ | $\mathbf{0.708}$ $\pm 5e^{-3}$ | $\mathbf{0.921}$ $\pm 1e^{-2}$ | $\mathbf{0.444}$ $\pm 1e^{-1}$ | $0.551$ $\pm 6e^{-2}$ | $\mathbf{0.966}$ $\pm 2e^{-3}$ |
| **Gradient Boosted Decision Trees (GBDT)** | | | | | | | |
| XGBoost | $\underline{0.872}$ $\pm 5e^{-4}$ | $0.375$ $\pm 1.2e^{-3}$ | $0.721$ $\pm 1e^{-3}$ | $0.951$ $\pm 1e^{-3}$ | $0.433$ $\pm 2e^{-3}$ | $\underline{0.729}$ $\pm 1e^{-3}$ | $0.980$ $\pm 1e^{-3}$ |
| CatBoost | $0.873$ $\pm 1e^{-3}$ | $0.381$ $\pm 1e^{-3}$ | $0.721$ $\pm 1e^{-3}$ | $0.946$ $\pm 9e^{-4}$ | $0.430$ $\pm 7e^{-4}$ | $0.726$ $\pm 8e^{-4}$ | $0.972$ $\pm 3e^{-3}$ |

tabular deep learning tasks such as binary and multi-class classification, as well as regression. We use as performance metrics Accuracy (↑) and RMSE (↓) for classification and regression respectively. The datasets we include in our experiments are Adult (AD) (Kohavi et al., 1996), Higgs (HI) (Vanschoren et al., 2014), Helena (HE) (Guyon et al., 2019), Jannis (JA) (Guyon et al., 2019), ALOI (AL) (Geusebroek et al., 2005) and California housing (CA) (Pace and Barry, 1997). We also add MNIST (interpreted as a tabular data) to our benchmark following Yoon et al. (2020). We summarize the characteristics of all 7 datasets in Table 11 in appendix C.

**Baselines**  To assess whether our self-supervised approach can foster performance on tabular applications, we compare the performance of several widely used tabular approaches with and without our self-supervised pre-training. We test our method on MLP (Taud and Mas, 2018), DCNV2 (Wang et al., 2021a), ResNet (He et al., 2016), AutoInt (Song et al., 2019) and FT-Transformer (Gorishniy et al., 2021). Similarly, as often considered as the go-to methods for supervised tasks on tabular data, we also compare the performance of T-JEPA augmented models to XGBoost (Chen and Guestrin, 2016) and CatBoost (Prokhorenkova et al., 2018). To further assess the relevance of T-JEPA to foster performance on supervised tasks, we compare the performance of MLP and ResNet when trained on the original dataspace representations and those generated by PTaRL (Ye et al., 2024), SwitchTab (Wu et al., 2024), BinRecon (Lee et al., 2024), SubTab (Ucar et al., 2021) VIME (Yoon et al., 2020) and T-JEPA.

**T-JEPA Training**  We split each dataset into training/validation/test sets (80/10/10) which were used for selecting both the hyperparameters of T-JEPA and of the models used for the downstream task. More precisely, in a model-agnostic manner, we relied on a systematic approach to train and evaluate the embedding space generated by T-JEPA. First, following previous work (Assran et al., 2023; Bardes et al., 2022) T-JEPA was trained on the training set and we conducted a hyperparameter tuning using a linear probe on the validation set to select the best configuration for each dataset. Second, we used the trained context encoder to generate data representations on which the subsequent model were trained. We refer the reader to appendix A.2 for more details on hyperparameter selection. Every experiment can be reproduced with the code provided in the supplementary material.

**Downstream task**  To adapt the T-JEPA representations to each model's input dimensions, we added a projection layer. We experimented with several techniques, including linear flattening, linear per-feature transformation, convolutional projections, and max and mean pooling. We refer the reader to appendix A.3 for more details. The projection layer was jointly trained with the downstream task and tailored to each model. Hyperparameter tuning was performed based on validation set

Table 2: **Comparison of different SSL models with ResNet and MLP as downstream model.** See Appendix F for more detail on experimental setting. We report and average over 20 runs and the corresponding standard deviation. The last columns displays the average rank over each dataset, the lower the better.

| | AD ↑ | HE ↑ | JA ↑ | AL ↑ | CA ↓ | HI ↑ | MNIST ↑ | Avg Rank ↓ |
|---|---|---|---|---|---|---|---|---|
| MLP | $0.827_{\pm 1e^{-3}}$ | $0.353_{\pm 1e^{-3}}$ | $0.672_{\pm 1e^{-3}}$ | $0.916_{\pm 4e^{-3}}$ | $0.511_{\pm 3e^{-3}}$ | $0.681_{\pm 4e^{-3}}$ | $0.978_{\pm 2e^{-3}}$ | 9.4 |
| +PTaRL | $\mathbf{0.868}_{\pm 4e^{-3}}$ | $0.383_{\pm 2e^{-3}}$ | $0.710_{\pm 4e^{-3}}$ | $0.917_{\pm 3e^{-3}}$ | $0.489_{\pm 2e^{-3}}$ | $0.713_{\pm 2e^{-3}}$ | $0.977_{\pm 1e^{-3}}$ | 5.1 |
| +SwitchTab | $0.867_{\pm N/A}$ | $0.387_{\pm N/A}$ | $0.726_{\pm N/A}$ | $0.942_{\pm N/A}$ | $0.452_{\pm N/A}$ | $\mathbf{0.724}_{\pm N/A}$ | N/A | 4.3 |
| +VIME | $0.859_{\pm 3e^{-3}}$ | $0.362_{\pm 6e^{-3}}$ | $0.695_{\pm 5e^{-3}}$ | $0.925_{\pm 5e^{-3}}$ | $0.505_{\pm 4e^{-2}}$ | $0.655_{\pm 6e^{-3}}$ | $0.941_{\pm 2e^{-3}}$ | 7.4 |
| + BinRecon | $0.816_{\pm 8e^{-3}}$ | $0.346_{\pm 2e^{-2}}$ | $0.581_{\pm 8e^{-2}}$ | $0.845_{\pm 5e^{-2}}$ | $0.498_{\pm 1e^{-2}}$ | $0.606_{\pm 8e^{-2}}$ | $0.918_{\pm 5e^{-2}}$ | 11.7 |
| + SubTab | $0.823_{\pm 3e^{-3}}$ | $0.378_{\pm 2e^{-3}}$ | $0.702_{\pm 1e^{-3}}$ | $0.954_{\pm 2e^{-3}}$ | $0.519_{\pm 3e^{-2}}$ | $0.673_{\pm 2e^{-3}}$ | $0.977_{\pm 2e^{-3}}$ | 8.1 |
| **+T-JEPA** | $0.866_{\pm 2e^{-3}}$ | $0.400_{\pm 4e^{-3}}$ | $\mathbf{0.728}_{\pm 3e^{-3}}$ | $0.961_{\pm 6e^{-3}}$ | $0.468_{\pm 4e^{-3}}$ | $0.517_{\pm 8e^{-2}}$ | $\mathbf{0.983}_{\pm 1e^{-3}}$ | 3.9 |
| ResNet | $0.814_{\pm 7e^{-3}}$ | $0.351_{\pm 2e^{-3}}$ | $0.666_{\pm 3e^{-3}}$ | $0.919_{\pm 2e^{-3}}$ | $0.534_{\pm 2e^{-3}}$ | $0.674_{\pm 4e^{-3}}$ | $0.979_{\pm 1e^{-3}}$ | 10.4 |
| +PTaRL | $0.862_{\pm 5e^{-3}}$ | $0.383_{\pm 2e^{-3}}$ | $0.723_{\pm 5e^{-3}}$ | $0.895_{\pm 1e^{-3}}$ | $0.498_{\pm 1e^{-3}}$ | $0.713_{\pm 2e^{-3}}$ | $0.973_{\pm 1e^{-3}}$ | 6.1 |
| + VIME | $0.851_{\pm 1e^{-3}}$ | $0.372_{\pm 2e^{-3}}$ | $0.699_{\pm 3e^{-3}}$ | $0.959_{\pm 1e^{-3}}$ | $0.505_{\pm 1e^{-3}}$ | $0.688_{\pm 3e^{-3}}$ | $0.933_{\pm 1e^{-3}}$ | 7.6 |
| + BinRecon | $0.828_{\pm 9e^{-3}}$ | $0.327_{\pm 1e^{-2}}$ | $0.699_{\pm 3e^{-3}}$ | $0.944_{\pm 1e^{-3}}$ | $0.471_{\pm 1e^{-2}}$ | $0.711_{\pm 3e^{-3}}$ | $0.981_{\pm 2e^{-3}}$ | 6.9 |
| + SubTab | $0.823_{\pm 3e^{-3}}$ | $0.365_{\pm 3e^{-3}}$ | $0.702_{\pm 1e^{-3}}$ | $0.958_{\pm 1e^{-3}}$ | $0.487_{\pm 2e^{-2}}$ | $0.675_{\pm 3e^{-3}}$ | $\mathbf{0.984}_{\pm 6e^{-4}}$ | 6.3 |
| **+T-JEPA** | $0.865_{\pm 3e^{-3}}$ | $\mathbf{0.401}_{\pm 2e^{-3}}$ | $0.718_{\pm 3e^{-3}}$ | $\mathbf{0.964}_{\pm 1e^{-3}}$ | $\mathbf{0.441}_{\pm 8e^{-2}}$ | $0.705_{\pm 5e^{-3}}$ | $0.983_{\pm 2e^{-3}}$ | **2.6** |

performance, and final evaluations were conducted on the test set, comparing results to models trained on the original dataset representations. To ensure the robustness and reliability of our findings, we conducted the experiment 20 times. We report the mean and standard deviation of the performance metrics. We refer the reader to appendix B for details on the downstream models' hyperparameters.

## 4.2 RESULTS

**Downstream Models** As displayed in Table 1, T-JEPA pre-training improved the performance of all evaluated models except on the Higgs (HI) dataset where only the performance of ResNet improves when augmented by T-JEPA. Interestingly, despite their design advantages for tabular data, recent attention-based models such as FT-Transformer and AutoInt, though improved by T-JEPA, did not surpass architectures like ResNet and MLP, which exhibited the most significant gains with T-JEPA. This outcome suggests that T-JEPA may complement the inductive biases of more traditional architectures like ResNet and MLP, which, despite their simpler design, are better equipped to leverage the feature representations learned through T-JEPA. Interestingly, when compared to gradient boosted decision trees, we observe that augmented with T-JEPA representation, the deep methods regularly obtain the best performance.

**Comparison to SSL methods** As displayed in Table 2, when compared to existing SSL methods, ResNet+T-JEPA obtains the lowest average rank while MLP+T-JEPA obtains the second lowest average rank. This emphasizes the relevance of T-JEPA for generating representation that foster performance on both regression and classification tasks. Other best performing alternative include MLP+SwitchTab and MLP+PTaRL that obtain respectively the third and fourth lowest average rank.

Overall these experiments demonstrate that T-JEPA contributes to increasing the performance of a wide range of downstream models. Moreover, it appears that T-JEPA outperforms existing SSL methods for tabular data when tested with MLP and ResNet as the downstream model.

## 5 DISCUSSION

### 5.1 REPRESENTATION SPACE EVALUATION

Two critical properties are considered desirable in a representation space: *uniformity* and *alignment* (Wang and Isola, 2020). Uniformity measures how well information contained in the original data space is preserved. Alignment describes a representation space in which semantically close samples are close to one another, while different samples should be distant. An edge case where both uniformity and alignment are not achieved is the one of representation collapse (see section 5.2) that describes a situation in which all samples are mapped to the same representation in the latent space.

**Metrics** We propose two complementary metrics to measure whether our representations are collapsed or satisfy the *uniformity* and *alignment* properties. First, to measure distribution consistency

Table 3: Key metrics tracking T-JEPA's representation learning on the Jannis (JA) dataset over different training epochs.

| Metric/Epoch | 0 | 20 | 40 | 60 | 80 | 100 | 120 |
|---|---|---|---|---|---|---|---|
| $D_{KL}$ | $9.30e^{-4}$ | $3.61e^{-4}$ | $6.62e^{-2}$ | $5.06e^{-2}$ | $6.55e^{-2}$ | $7.66e^{-2}$ | $9.38e^{-2}$ |
| $\|\cdot\|_2$ | 5.83 | 3.93 | 54.3 | 50.6 | 68.1 | 71.4 | 70.0 |
| uniformity | 3.12 | 0.94 | 10.69 | 11.20 | 11.23 | 11.32 | 11.38 |

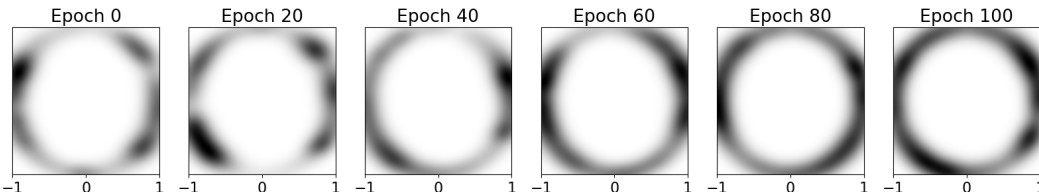

Figure 2: **Visualization of the representation space** at various epochs of the T-JEPA pretraining on the Jannis (JA) dataset. Each plot depicts the density of transformed points in two dimensions, with darker areas indicating higher density.

and alignment, we rely on the Kullback-Leibler divergence ($D_{KL}(\uparrow)$): collapsed representations would imply similar distributions between diverse samples in the embedding space. Thus, lower values would indicate collapse as different samples have indistinct distributions. In contrast, higher values show that dissimilar samples tend to be located farther apart. We expect $D_{KL}$ to increase as training progresses. Second, we rely on the uniformity($\uparrow$) metric (Wang and Isola, 2020) to measure how much of the representation space is utilized. Collapsed representations would imply that samples are tightly clustered in a small region of the latent space, achieving low uniformity. On the contrary, desirable representations are more uniformly spread out, effectively using the embedding space. See Appendix G for more detail on the metrics.

**Embedding space characterization**  To assess whether T-JEPA generates representations uniformly spanned across the embedding space, we display in Figure 2 the obtained sample representations for the JA dataset at different training epochs. We randomly sample 50,000 points and rely on the PaCMAP (Wang et al., 2021b) dimensionality reduction technique to reduce their representations in the embedding space to two dimensions for visualization purposes. We observe in Figure 2 that T-JEPA effectively utilizes the representation space, evolving from an initial collapsed distribution towards a more structured arrangement as the training progresses. This behavior is indicative of the model's capacity to learn distinct and meaningful representations, which are crucial for downstream tasks. We also display in Table 3 how the uniformity score, KL divergence ($D_{KL}$), and the euclidean distance ($\|\cdot\|_2$) vary during training. At each epoch, we randomly select 50,000 sample representations from the JA dataset and compute these metrics. For both KL-Divergence and euclidean distance, we report the average pairwise KL-Divergence and euclidean distance for the 50,000 samples previously selected. The increasing KL divergence and euclidean distance demonstrate that the model effectively avoids representation collapse and better satisfies the *alignment* property as training progresses. Indeed, T-JEPA's training objective pushes samples farther from one another which facilitates discrimination between samples for supervised tasks. Finally, increasing uniformity across epochs is in line with the representations displayed in Figure 2 and demonstrates that the training objective preserves information from the original dataspace.

## 5.2 REPRESENTATION COLLAPSE

As a non-contrastive self-supervised learning method, T-JEPA is prone to representation collapse as discussed in previous work (Bardes et al., 2024; Assran et al., 2023). EMA combined with a stop-gradient operation has been considered to be sufficient to prevent JEPA-based methods from leading to degenerate solutions in which all samples have the same representation. However, we observed that further regularization of T-JEPA was necessary to avoid such pitfall. Specifically,

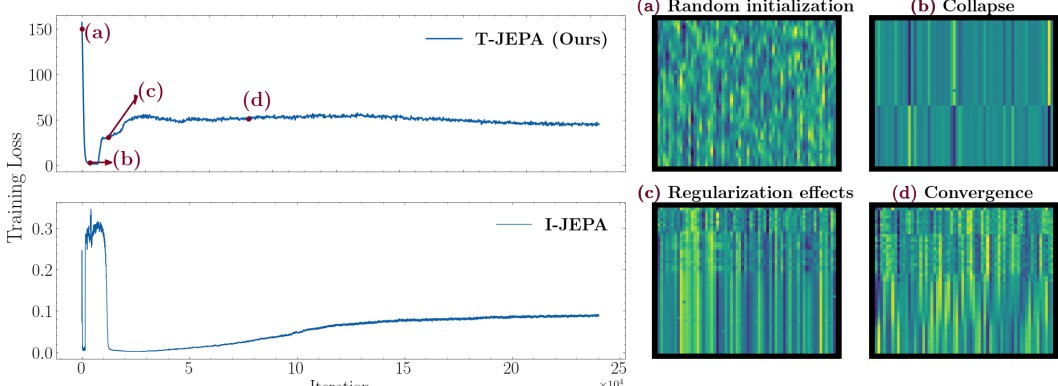

Figure 3: **Training regime** of Joint-Embedding Predictive Architectures on tabular (JA) and image (ImageNet-1K) data. We display on the right a randomly selected sample's representations for each critical part of the training process. The subfigures (a) to (d) illustrate the evolving outputs of the context encoder $h_{\text{context}} \in \mathbb{R}^{d \times h}$. In each heat-map, rows correspond to the $d$ features, while columns represent the $h$ hidden dimensions. (a) describes the initial random initialization, (b) the collapsed equilibrium, (c) the regularization effect pushing the weights outside of the collapsed equilibrium, and (d) the convergence.

appending a regularization token `[REG]` to both target and context representations appeared critical to escape the initial representation collapse.

**Initial Collapse**   We observe that JEPA-based models undergo an initial representation collapse due to the EMA relation between the weights of the context and label encoders. Indeed, we retrained from scratch I-JEPA (Assran et al., 2023) on ImageNet-1K using their official implementation and hyperparameters[1] and observed a similar training regime as for T-JEPA. First, the loss collapses close to 0 after a few iterations; then, regularization starts pushing the model's weights toward a non-trivial equilibrium. We display in Figure 3 the training regimes of both T-JEPA and I-JEPA.

**Regularization Token**   While other JEPA methods do not appear to require further regularization, we observe that appending a regularization token `[REG]` to the sample's representations is instrumental in escaping training collapse regimes. As displayed in Figure 4, when training T-JEPA on the Jannis dataset without including any regularization token, the optimization process does not manage to escape the initial collapse.

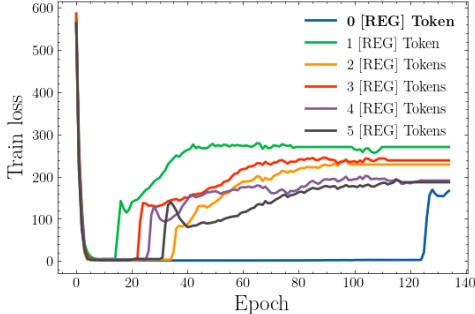

Figure 4: **Regularization token ablation**. Training loss for the JA dataset across different numbers of regularization tokens `[REG]`.

On the contrary, when including one or more tokens, the optimization process can escape this initial collapse and pushes the weights towards a non-trivial equilibrium.

### 5.3   COMPARISON WITH GRADIENT BOOSTED DECISION TREES

**Performance Comparison**   Recent work (Grinsztajn et al., 2022; Gorishniy et al., 2024) discusses how neural networks tend to struggle with structured tabular data type in comparison with other non-deep methods based on gradient-boosted decision trees (GBDT). In most scenarios, approaches such as XGBoost (Chen and Guestrin, 2016) or CatBoost (Prokhorenkova et al., 2018) surpass deep learning algorithms. We observe in Table 1 that on most datasets, methods trained on raw data are

---

[1]For computational purposes, we reduced the batch size to 16 and kept unchanged the rest of the hyperparameters.

significantly outperformed by both GBDT methods. However, once augmented by T-JEPA, most methods consistently outperform GBDT or match their performance.

**Feature Importance** To try and understand why T-JEPA enabled some approaches to match the performance of GBDT, we investigated whether high-variance features in the latent space correlate with feature importance for downstream tasks. While the $d$ representations do not exactly correspond to a one-to-one relation with the corresponding feature in the original dataspace, index embeddings allow the obtained encoded representations to still hold feature-related information. Including index embeddings does not eliminate token mixing from residual/self-attention modules, nevertheless it still helps maintain a degree of alignment between features and representations. The embedding variance was computed as the standard deviation of each feature's values across hidden dimensions, normalized by the dimension-wise mean. We ranked features by their average embedding variance across all samples and compared these rankings to those generated by traditional supervised methods, including XGBoost and permutation importance, using Kendall's $\tau$ for rank similarity. Overall, our analysis reveals a strong correlation between high-variance features in the T-JEPA embeddings and those identified as important by supervised methods, despite T-JEPA being trained without target labels. Figure 5 provides a detailed comparison. The embedding variance ranking ($\sigma_{\text{embed}}$) shows significant correlation with XGBoost ($\tau = 0.44$, with $p$-value $= 1.73e^{-6}$). A random generated rank is provided for comparison. These findings suggest that T-JEPA's self-supervised framework effectively captures key features relevant to downstream tasks without supervision. The alignment between embedding variance and traditional feature importance further highlights the model's ability to learn meaningful data representations.

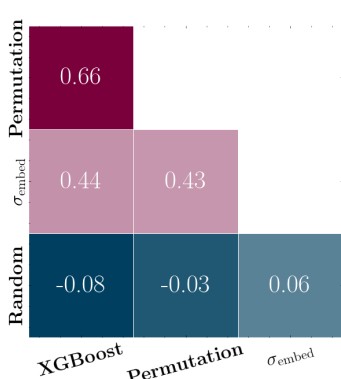

Figure 5: Pairwise comparison of feature rankings using Kendall's $\tau$ correlation on the JA dataset. Rankings are derived from XGBoost feature importance, permutation importance, T-JEPA's embedding variance ($\sigma_{embed}$), and a random baseline.

## 6 Conclusion

Overall, we have proposed a novel augmentation-free self-supervised method for representation learning that has demonstrated strong performance across both classification and regression tasks on diverse datasets. We have investigated the characteristics of the obtained representations and demonstrated that our approach is relevant as it identifies pertinent features without access to the downstream task's target. Moreover, most methods augmented by T-JEPA outperform or match the performance of GBDT, which is often considered the go-to method for supervised tasks on tabular data. In particular, aligned with previous work that demonstrated that ResNet was a solid alternative to GBDT for tabular data (Gorishniy et al., 2021; Zabërgja et al., 2024), we observe that ResNet+T-JEPA outperforms all competing methods on most datasets, including GBDT. Finally, we empirically uncovered a novel regularization method for transformers on tabular data by including a regularization token that prevents from entering collapsed training regimes.

**Limitations and Future Work** Our method generates representations best suited for transformer-like architecture that requires to be *adapted* to other architectures. Future work may include investigating other approaches to adapting JEPA-like methods for representation learning of tabular data that would be suited for other architectures than transformers. Other possible extensions of the present work might include investigating using JEPA-like reconstruction as a pretext task for self-supervised anomaly detection on tabular data. Finally, our work has emphasized the uncanny training regimes of JEPA-like methods as non-contrastive self-supervised approaches. Further investigation to provide better theoretical insight into why non-contrastive self-supervised learning works well might enable the construction of novel designs and approaches that foster performance.

**Acknowledgement**   This work was performed using HPC resources from the "Mésocentre" computing center of CentraleSupélec and École Normale Supérieure Paris-Saclay supported by CNRS and Région Île-de-France (http://mesocentre.centralesupelec.fr/). This work was also granted access to the HPC resources of IDRIS under the allocation 2024-101424 made by GENCI. This research publication is supported by the Chair "Artificial intelligence applied to credit card fraud detection and automated trading" led by CentraleSupelec and sponsored by the LUSIS company.

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

# A   T-JEPA TRAINING SETTINGS

This section presents the implementation details of the project.

## A.1   PROGRAMMING ENVIRONMENT

The code environment for this project was implemented using Python and several third-party libraries. Table 4 details the main libraries used along with their respective versions. The training was done on a single NVIDIA HGX A100 GPU with 40GB of memory.

Table 4: Main libraries used in the project.

| Library | Description |
|---|---|
| `Python` v3.12.2 | The programming language used for the project |
| `einops` v0.8.0 | A flexible and powerful tool for tensor operations |
| `matplotlib` v3.8.4 | A library for creating static, animated, and interactive plots |
| `numpy` v2.1.0 | Fundamental package for scientific computing with arrays |
| `pandas` v2.2.2 | Data manipulation and analysis tool |
| `pytorch_lightning` v2.2.1 | A PyTorch wrapper for high-performance deep learning research |
| `scikit_learn` v1.4.1.post1 | Machine learning library for Python |
| `scipy` v1.14.1 | Library for scientific and technical computing |
| `torch` v2.3.0.post301 | PyTorch deep learning library |
| `torchinfo` v1.8.0 | Module to show model summaries in PyTorch |
| `tqdm` v4.66.2 | Progress bar utility for Python |
| `xgboost` v2.1.1 | Optimized gradient boosting library |

## A.2   HYPERPARAMETER SEARCH

We employed Bayesian optimization to tune the hyperparameters of T-JEPA. The batch size was fixed at $512$ for all configurations, while the exponential moving average (EMA) decay rate was set to vary from $0.996$ to $1$. Additionally, we used four prediction masks throughout the training process. For optimization, we selected the AdamW optimizer (Loshchilov and Hutter, 2019) due to its proven robustness in large-scale models. The learning rate was adaptively adjusted using a cosine annealing scheduler (Loshchilov and Hutter, 2017), which gradually reduced it from the initial value to a minimum, $\eta_{\min} = 0$.

Table 5: Hyperparameter Configuration for Bayesian Optimization

| Parameter | Values |
|---|---|
| `model_num_heads` | $[2, 4, 8]$ |
| `model_dim_hidden` | $[2, 4, 8, 16, 32, 64, 128]$ |
| `model_num_layers` | $[1, 2, 3, 4, 5, 6, 7, 8, 16]$ |
| `model_dim_feedforward` | $[64, 128, 256, 512, 768, 1024]$ |
| `model_dropout_prob` | $(0.0, 0.01)$ |
| `exp_lr` | $(0.00001, 0.001)$ |
| `mask_min_ctx_share` | $(0.07, 0.15)$ |
| `mask_max_ctx_share` | $(0.2, 0.9)$ |
| `mask_min_trgt_share` | $(0.05, 0.20)$ |
| `mask_max_trgt_share` | $(0.2, 0.9)$ |
| `pred_num_layers` | $[2, 4, 8, 16, 24, 32]$ |
| `pred_embed_dim` | $[4, 8, 16, 32, 64, 128]$ |
| `pred_num_heads` | $[2, 4, 8]$ |
| `pred_p_dropout` | $(0.0, 0.01)$ |

The hyperparameters displayed in table 5 correspond to the following:

- `model_num_heads`: number of attention heads of the context encoder $f_\theta$.
- `model_dim_hidden`: hidden dimension of the context encoder $f_\theta$.
- `model_num_layers`: number of layers of the context encoder $f_\theta$.
- `model_dim_feedforward`: dimension of FFN in the context encoder $f_\theta$.
- `model_dropout_prob`: dropout probability of the context encoder $f_\theta$.
- `exp_lr`: learning rate.
- `mask_min_ctx_share`: Minimum share of masked feature for the context representation (see (a) in Figure 1).
- `mask_max_ctx_share`: Maximum share of masked feature for the context representation (see (a) in Figure 1).
- `mask_min_trgt_share`: Minimum share of masked feature for the target representation (see (b) in Figure 1).
- `mask_max_trgt_share`: Maximum share of masked feature for the target representation (see (b) in Figure 1).
- `pred_num_heads`: number of attention heads of the predictor $g_\phi$.
- `pred_num_layers`: number of layers of the predictor $g_\phi$.
- `pred_embed_dim`: hidden dimension of the predictor $g_\phi$.
- `pred_p_dropout`: dropout probability of the predictor $g_\phi$.

### A.3 PROJECTION LAYER

The projection layer adapts the T-JEPA representation space $h \in \mathbb{R}^{d \times h}$ to the input dimensions required by the downstream models. Several projection techniques were implemented, as described below:

- **Linear Flatten:** When using the linear flatten projection, the input is flattened into a single vector $\mathbf{h}_{\text{flatten}} \in \mathbb{R}^{d \cdot h \times 1}$ and transformed through a linear projection, $\mathbf{h}_{\text{proj}} = \mathbf{W} \cdot \mathbf{h}_{\text{flatten}} + \mathbf{b} \in \mathbb{R}^{h_{\text{new}} \times 1}$, with $\mathbf{W} \in \mathbb{R}^{h_{\text{new}} \times d \cdot h}$ the weight matrix and $\mathbf{b} \in \mathbb{R}^{h_{\text{new}} \times 1}$ the bias.

- **Linear Per-Feature:** Each feature is transformed independently by applying a linear projection to each feature vector $\mathbf{h}_i \in \mathbb{R}^{d \times 1}$, $\mathbf{h}_{i,\text{proj}} = \mathbf{W}_i \cdot \mathbf{h}_i + \mathbf{b}_i \in \mathbb{R}$, where $\mathbf{W}_i \in \mathbb{R}^{1 \times d}$ is the weight matrix and $\mathbf{b}_i \in \mathbb{R}$ is the bias for each feature $i$.

- **Convolutional Projection:** The convolutional encoder applies two stages of convolution followed by max pooling. For an input representation $\mathbf{x} \in \mathbb{R}^{d \times h}$, where $d$ is the number of feature and $h$ is the hidden dimension, the convolution operation is represented as: $\mathbf{x}' = \sigma\left(\text{BatchNorm}(\text{Conv2D}(\mathbf{X}, \mathbf{k}_1))\right)$, where $\mathbf{k}_1$ is a convolutional kernel, and $\sigma$ is the activation function (ReLU). After a second convolution and pooling step, the final representation is flattened into a vector and projected to the target embedding dimension.

- **Max Pooling:** The max pooling operation selects the maximum value for each feature vector $\mathbf{h}_i \in \mathbb{R}^h$: $\mathbf{h}_{i,\max} = \max(\mathbf{h}_{i,j} \mid j \in [d])$.

- **Mean Pooling:** The mean pooling operation averages values for each feature vector $\mathbf{h}_i \in \mathbb{R}^h$: $\mathbf{h}_{i,\text{mean}} = \frac{1}{h} \sum_{j \in [h]} \mathbf{h}_{i,j}$.

Each projection layer is trained jointly with the downstream task and is selected based on the structure of the input data and model requirements.

## B DOWNSTREAM MODELS' ARCHITECTURE

Similarly to T-JEPA, we employed Bayesian optimization to tune the hyperparameters of the downstream model's architecture. We give details on each architecture and their corresponding hyperparameters in the present section.

## B.1 MLP

The Multi-Layer Perceptron (MLP) architecture employed in this work is designed to effectively transform input features. The transformation begins with a linear projection of the input $\mathbf{z}$, which can either be the raw input $\mathbf{x}$ or an embedding projection, into a hidden representation of dimensionality $h$:

$$\mathbf{h}^{(0)} = \mathbf{W}_1 \mathbf{z} + \mathbf{b}_1.$$

The model then applies the hidden layers sequentially:

$$\mathbf{h}^{(l+1)} = \text{BatchNorm}\big(\text{Dropout}\big(\text{ReLU}\big(\mathbf{W}_{l+1}\mathbf{h}^{(l)} + \mathbf{b}_{l+1}\big)\big)\big),$$

where $l = 0, \ldots, L - 1$. Here is the selected hyper-parameters for each dataset:

Table 6: Hyperparameters of MLP Model for Each Dataset

| Dataset | Dropout | Encoder Type | Learning Rate | Weight Decay | Hidden Layers |
|---------|---------|--------------|---------------|--------------|---------------|
| HE | 0.5478 | `linear_flatten` | $8.97 \times 10^{-2}$ | $4.45 \times 10^{-4}$ | 4 |
| HI | 0.4203 | `linear_per_feature` | $1.84 \times 10^{-5}$ | $7.48 \times 10^{-4}$ | 9 |
| JA | 0.4923 | `linear_per_feature` | $7.31 \times 10^{-4}$ | $2.09 \times 10^{-6}$ | 13 |
| AD | 0.2311 | `linear_per_feature` | $1.35 \times 10^{-4}$ | $6.43 \times 10^{-4}$ | 13 |
| CA | 0.0310 | `linear_flatten` | $1.19 \times 10^{-5}$ | $1.96 \times 10^{-5}$ | 3 |
| AL | 0.2331 | `linear_flatten` | $4.87 \times 10^{-4}$ | $1.38 \times 10^{-4}$ | 4 |
| MNIST | 0.3527 | `linear_flatten` | $1.83 \times 10^{-5}$ | $1.47 \times 10^{-4}$ | 5 |

## B.2 IMPROVED DEEP CROSS NETWORK

This section presents the enhanced `DCN-V2` model architecture, designed to learn both explicit and implicit feature interactions. The code used in this work was taken from (Wang et al., 2021a). The `DCN-V2` model combines a Cross Network with a Deep Network, achieving superior expressiveness while maintaining computational efficiency. The explicit feature interactions are modeled through the Cross Network layers, defined as:

$$\mathbf{x}_{l+1} = \mathbf{x}_0 \odot (\mathbf{W}_l \mathbf{x}_l + \mathbf{b}_l) + \mathbf{x}_l, \quad \mathbf{W}_l \in \mathbb{R}^{d \times d}, \ \mathbf{b}_l \in \mathbb{R}^d.$$

Here, $\mathbf{x}_0$ represents the base features, and $\odot$ denotes element-wise multiplication. Implicit feature interactions are captured by a Deep Network, where the $l$-th layer is defined as:

$$\mathbf{h}_{l+1} = f(\mathbf{W}_l \mathbf{h}_l + \mathbf{b}_l), \quad f(\cdot) = \text{ReLU}(\cdot).$$

Table 7 summarizes the hyperparameters used in experiments.

Table 7: Hyperparameters for `DCN-V2` experiments.

| Dataset | Cross Dropout | Embedding Dim. | Hidden Dim. | Hidden Dropout | Learning Rate | Weight Decay |
|---------|---------------|----------------|-------------|----------------|---------------|--------------|
| HE | 0.0808 | 128 | 768 | 0.2926 | $8.81 \times 10^{-5}$ | $9.33 \times 10^{-4}$ |
| HI | 0.0346 | 7 | 488 | 0.0879 | $5.53 \times 10^{-4}$ | $3.64 \times 10^{-4}$ |
| JA | 0.0702 | 70 | 386 | 0.0862 | $8.79 \times 10^{-5}$ | $1.21 \times 10^{-7}$ |
| AD | 0.1393 | 45 | 428 | 0.2976 | $4.69 \times 10^{-5}$ | $1.21 \times 10^{-4}$ |
| CA | 0.2578 | 66 | 704 | 0.1111 | $4.91 \times 10^{-5}$ | $1.11 \times 10^{-5}$ |
| AL | 0.1476 | 25 | 524 | 0.0431 | $1.94 \times 10^{-5}$ | $1.40 \times 10^{-6}$ |
| MNIST | 0.2836 | 93 | 755 | 0.0875 | $4.22 \times 10^{-5}$ | $1.77 \times 10^{-5}$ |

## B.3 RESNET

The ResNet model used in this work was specially tailored for tabular data. The model is formalized as follows:

$$\text{ResNet}(x) = \text{Prediction}(\text{ResNetBlock}(\ldots(\text{ResNetBlock}(\text{Linear}(x)))))$$
$$\text{ResNetBlock}(x) = x + \text{Dropout}(\text{Linear}(\text{Dropout}(\text{ReLU}(\text{Linear}(\text{BatchNorm}(x)))))) \quad (2)$$
$$\text{Prediction}(x) = \text{Linear}(\text{ReLU}(\text{BatchNorm}(x)))$$

Table 8: Hyperparameters for ResNet Variants

| Dataset | d_block | dropout1 | dropout2 | lr | n_blocks |
|---------|---------|----------|----------|-----|----------|
| HE | 512 | 0.459 | 0.461 | $2.74 \times 10^{-4}$ | 2 |
| AD | 440 | 0.042 | 0.137 | $8.90 \times 10^{-4}$ | 5 |
| CA | 465 | 0.313 | 0.002 | $5.26 \times 10^{-5}$ | 4 |
| HI | 506 | 0.234 | 0.031 | $2.24 \times 10^{-5}$ | 2 |
| AL | 476 | 0.044 | 0.049 | $1.32 \times 10^{-5}$ | 8 |
| JA | 355 | 0.137 | 0.005 | $4.60 \times 10^{-5}$ | 7 |
| MNIST | 497 | 0.265 | 0.050 | $6.25 \times 10^{-4}$ | 4 |

## B.4 AUTOINT

AutoInt is a neural network leveraging self-attention mechanisms for automatic feature interaction learning. The input features $\mathbf{x} \in \mathbb{R}^n$ are embedded into vectors $\mathbf{e}_i \in \mathbb{R}^d$ that interact through multi-head attention. These interactions produce interaction-adjusted embeddings $\tilde{\mathbf{e}}_i$. A residual connection ensures the retention of low-order information:

$$\mathbf{e}_i^{\text{Res}} = \text{ReLU}(\tilde{\mathbf{e}}_i + \mathbf{W}_{\text{Res}}\mathbf{e}_i).$$

The final prediction is computed as:

$$\hat{y} = \sigma(\mathbf{w}^\top[\mathbf{e}_1^{\text{Res}} \oplus \cdots \oplus \mathbf{e}_M^{\text{Res}}] + b),$$

where $\sigma(x)$ represents the sigmoid function.

The hyperparameter configurations for different datasets are summarized in Table 9.

Table 9: Hyperparameter configurations for the AutoInt model across datasets.

| Dataset | d_token | Num. layers | Learning rate | Residual Dropout | Weight Decay |
|---------|---------|-------------|---------------|------------------|--------------|
| AD | 200 | 1 | $1.717 \times 10^{-3}$ | 0.060 | $1.782 \times 10^{-7}$ |
| CA | 238 | 8 | $2.141 \times 10^{-4}$ | 0.071 | $1.892 \times 10^{-6}$ |
| HI | 166 | 1 | $6.913 \times 10^{-4}$ | 0.043 | $1.385 \times 10^{-7}$ |
| AL | 228 | 3 | $2.280 \times 10^{-3}$ | 0.011 | $4.667 \times 10^{-4}$ |
| JA | 32 | 6 | $9.680 \times 10^{-4}$ | 0.090 | $1.234 \times 10^{-4}$ |
| HE | 128 | 6 | $5.193 \times 10^{-5}$ | 0.055 | $7.590 \times 10^{-4}$ |
| MNIST | 252 | 2 | $5.831 \times 10^{-4}$ | 0.091 | $5.107 \times 10^{-5}$ |

## B.5 FT-TRANSFORMER

The FT-Transformer processes tabular data using three key stages: feature tokenization, sequential Transformer layers, and a prediction head. Feature tokenization encodes each numerical feature $x_j$ and categorical feature $e_j$ as:

$$T_j^{(\text{num})} = b_j^{(\text{num})} + x_j W_j^{(\text{num})}, \quad T_j^{(\text{cat})} = b_j^{(\text{cat})} + e_j W_j^{(\text{cat})}.$$

These embeddings are concatenated into a matrix $T \in \mathbb{R}^{k \times d}$, where $k$ is the number of features and $d$ is the embedding dimension. A [CLS] token is prepended to $T$, and $L$ Transformer layers are applied, iteratively updating the representation as follows:

$$T_i = \text{MHSA}(\text{LN}(T_{i-1})) + T_{i-1}, \quad T_i = \text{FFN}(\text{LN}(T_i)) + T_i,$$

where LN is layer normalization. The prediction head computes the final output by processing the [CLS] token with a normalized and activated linear transformation:

$$\hat{y} = W_2 \cdot \text{ReLU}(W_1 \cdot \text{LayerNorm}(T_L^{[\text{CLS}]})).$$

The following table summarizes the hyperparameters for the `FT-Transformer` model applied to various datasets.

Table 10: Hyperparameter Configuration per Dataset

| Dataset | Att. Dropout | Num. Heads | Block Size | Hidden Dim | Learning Rate |
|---------|-------------|-----------|-----------|-----------|--------------|
| HE | 0.0157 | 16 | 64 | 128 | $1.08 \times 10^{-3}$ |
| HI | 0.3334 | 4 | 128 | 64 | $2.25 \times 10^{-4}$ |
| JA | 0.0934 | 16 | 64 | 256 | $1.33 \times 10^{-4}$ |
| AD | 0.3426 | 16 | 128 | 32 | $1.35 \times 10^{-4}$ |
| CA | 0.0745 | 8 | 128 | 512 | $1.17 \times 10^{-3}$ |
| AL | 0.3246 | 8 | 256 | 64 | $8.27 \times 10^{-4}$ |
| MNIST | 0.4259 | 8 | 32 | 256 | $7.97 \times 10^{-5}$ |

## C  COMPUTE AND TRAINING TIME

The training process was executed on a single NVIDIA A100 GPU. Despite the varying dataset sizes, we aimed to optimize compute efficiency by carefully tuning the batch size and learning rate for each experiment. As detailed in table 11, the pretraining duration across datasets ranged from 0.34 GPU-hours to 3.64 GPU-hours. These variations largely reflect the complexity of the datasets in terms of both sample size and feature dimensions. In total, the computational demand remained within acceptable limits, allowing us to complete multiple runs with reasonable turnaround times, while also maintaining a balance between model performance and resource usage.

Table 11: Dataset characteristics and **pretraining GPU-hours.**

| | AD | HI | HE | JA | AL | CA | MNIST |
|---|---|---|---|---|---|---|---|
| Samples | 48,842 | 98,050 | 65,196 | 83,733 | 108,000 | 20,640 | 67112 |
| Numerical, Categorical | 6, 8 | 28, 0 | 27, 0 | 54, 0 | 128, 0 | 8, 0 | 784, 0 |
| Classes | 2 | 2 | 100 | 4 | 1,000 | N/A | 10 |
| Metric | Accuracy | Accuracy | Accuracy | Accuracy | Accuracy | RMSE | Accuracy |
| **Pretraining GPU-hours** | 0.84 | 1.80 | 1.03 | 1.27 | 3.64 | 0.34 | 4.80 |

## D  ALTERNATIVE STRATEGIES

Other settings have been considered before the one discussed in section 3, and were **discarded because they led to collapsed regimes**. We considered two alternatives. First, we considered a pipeline where only the masking strategy differs, as detailed in section D.1. Second, we also considered an alternative where the masking strategy is the one detailed in section D.1, and we also modify the predictor architecture as detailed in section D.2.

### D.1  MASKING STRATEGY

**Masking**  Following previous work on feature masking for tabular data, we considered handling masked features by keeping the sample's representation's dimension constant. Features of a samples are normalized similarly as detailed in section 3 such that $\mathbf{E}(\mathbf{x}_j) \in \mathbb{R}^{e_j}$, where $e_j = 1$ for numerical features and for categorical features $e_j$ corresponds to their cardinality. Each sample is accompanied by a masking vector $\mathbf{m} \in \{0,1\}^d$ in which each entry designates whether a feature is masked: $m_j = \mathbb{1}\{\text{feature } j \text{ is masked}\}$. When masked, we replace the corresponding feature value with $0$ and concatenate each feature representation with the corresponding mask indicator function. Hence, each feature $j$ has an $(e_j + 1)$-dimensional representation

$$\tilde{\mathbf{E}}(\mathbf{x}) = ((\mathbf{1}_d - \mathbf{m}) \odot \mathbf{E}(\mathbf{x}), \mathbf{m}) \in \mathbb{R}^{d \times (e_j + 1)}, \tag{6}$$

where $\odot$ designates the Hadamard product and $\mathbf{1}_d$ the $d$-dimensional unit vector.

We pass each of the $d$ features encoded representations of sample $\mathbf{x}$ through $d$ learned linear layers `Linear`$(e_j + 1, h)$. We also learn $h$-dimensional index and feature-type embeddings. Both are added to the embedded representation of sample $\mathbf{x}$. Let $\mathbf{z}_{\mathbf{x}}^{\mathbf{m}} \in \mathbb{R}^{d \times h}$ denote the obtained embedded representation of sample $\mathbf{x}$ with mask $\mathbf{m}$.

**Context and Target Encoders**   Given this modification, the obtained context representation's dimension differ from the one given in equation (2),

$$h^{\mathbf{m}}_{\text{context}} = f_\theta(\mathbf{z}^{\mathbf{m}}_{\mathbf{x}}) \in \mathbb{R}^{d \times h}. \quad \text{(context)}$$

The rest of the pipeline is identical to the one given in section 3. We also considered a second alternative where the above pipeline is chosen but a different predictor architecture (see section D.2) replaces the transformer as detailed in section 3.

### D.2   PREDICTOR

Given this alternative masking strategy, we also considered an MLP-based predictor for target representation prediction. The predictor consists of $d$ separate MLPs (one for each feature). Each MLP takes as input a representation of dimension $d \cdot h$ (the flattened representation output by the context encoder), and produces and output of dimension $h$. In particular, each MLP corresponds to one feature in particular and produces an $h$-dimensional prediction for the corresponding feature given a flattened context representation of dimension $d \cdot h$.

Let us denote $g_\phi = \{\text{MLP}_i\}^d_{i=1}$, a target mask $\mathbf{m}_{tgt}$ and $z_{flatten} = \texttt{flatten}(h^{\mathbf{m}}_{context})$. Then the prediction for the target mask $\mathbf{m}_{target}$ is given by,

$$\hat{h}^{\mathbf{m}_{target}}_{target} = \{MLP_j(z_{flatten}) : \mathbf{m}^j_{target} = 0\} \tag{7}$$

## E   EMBEDDING FEATURE VARIANCE

The idea behind calculating the variance of embedding features is based on the assumption that features with high variability across the embedding space are more expressive and likely to capture the underlying structure of the data. As presented in Figure 6, some features (rows in the heatmap) present more perturbations across the hidden dimensions (columns in the heatmap).

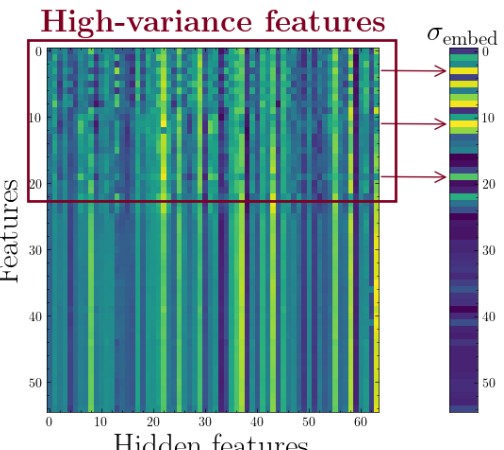

Figure 6: **Embedding Variance**

To measure this variance, let us define the embedding variance score. Let $\mathbf{x} \in \mathbb{R}^{d \times h}$ represent a point the latent space, where $d$ is the number of features and $h$ is the hidden dimension. For each hidden dimension $j$, we compute the mean $\mu_j$ across the $d$ features, i.e., $\mu_j = \frac{1}{d} \sum^d_{i=1} x_{i,j}$. For each feature $i$, we calculate the embedding variance score, defined as $\sigma_{i,\text{embed}} = \text{Var}(\mathbf{x_i} - \mu)$, where $\mathbf{x_i}$ is the vector representing the $i$-th-feature, and $\mu = [\mu_0, \ldots \mu_h]^T$. Features with higher embedding variance are those that stand out more in the representation space, suggesting they may carry more relevant information.

# F    EXPERIMENTAL SETTING

In Table 1, we present performance metrics for several models, including neural networks (MLP, DCNv2, ResNet, AutoInt, FT-Trans), and with and without T-JEPA. For both cases, we relied on bayesian optimization to find the best set of hyperparameters according to performance on a validation set. We report the performance obtained with this set of hyperparameters on a test set never used during training.

In Table 2 we report the performance of several SSL methods (PTaRL (Ye et al., 2024), SwitchTab (Wu et al., 2024), BinRecon (Lee et al., 2024) and SubTab (Ucar et al., 2021)). Metrics displayed in this table are obtained as follows:

- For PTaRL (Ye et al., 2024), we report the metrics from their paper for datasets AD, JA and CA as the authors kindly shared the standard deviations corresponding to these datasets for this model. For the remaining datasets (HE, AL, HI and MNIST) we display metrics obtained from our own experiments and their corresponding standard deviations.

- For BinRecon (Lee et al., 2024), metrics for MLP are primarily obtained from their paper, while we run the experiments ourselves for the ResNet alternative.

- For SwitchTab (Wu et al., 2024), metrics are obtained from their paper for the MLP dosnwtream model, except for MNIST as they did not use this dataset in their benchmark. Moreover, as the authors have not released any code for their method, we are unable to obtain any metric for the MNIST dataset and the ResNet downstream model. Moreover, no standard deviations are reported in their paper.

- For VIME (Yoon et al., 2020), we used the official code made available online by the authors to run the experiments for HE, JA, AL, CA and HI datasets and report the metrics from their paper for AD and MNIST for the MLP downstream model, and run the experiments for all datasets for the ResNet downstream model. Notably, we rely on the self-supervised only set-up for fair comparison with other methods.

- For SubTab (Ucar et al., 2021), we rely on their official implementation and run experiments on each of the datasets except for AD which we obtained from their paper for the MLP downstream model.

For each SSL model, downstream model and dataset we rely on bayesian optimization to set the hyperparameters. This ensure fair and reproducible comparison between all methods. We provide the code to replicate all our experiments on the official github repository and we display in appendix F.1 and F.2 the experimental details.

## F.1    BASELINE PERFORMANCE

The default hyperparameters for each model were utilized to ensure consistent configurations and establish baseline performance metrics. To maintain alignment with the experimental framework presented in Ye et al. (2024), the same hyperparameters were adopted for generating the baseline results.

Table 12: Default Hyperparameters for `MLP`

| Hyperparameter | Value |
|---|---|
| Number of Layers | 4 |
| Hidden Dimension | 256 |
| Dropout | 0.1 |
| Batch Size | 128 |
| Learning Rate | $1 \times 10^{-4}$ |
| Early Stopping Patience | 16 |
| Maximum Epochs | 200 |
| Categorical Embedding Dim | 128 |

Table 13: Default Hyperparameters for `DCNV2`

| Hyperparameter | Value |
|---|---|
| Hidden Dimension | 128 |
| Number of Cross Layers | 3 |
| Number of Hidden Layers | 7 |
| Cross Dropout | 0.1 |
| Hidden Dropout | 0.1 |
| Batch Size | 128 |
| Learning Rate | $1 \times 10^{-4}$ |
| Early Stopping Patience | 16 |
| Maximum Epochs | 200 |
| Categorical Embedding Dim | 128 |

Table 14: Default Hyperparameters for `ResNet`

| Hyperparameter | Value |
|---|---|
| Number of Layers | 4 |
| Hidden Dimension | 256 |
| Hidden Dropout | 0.1 |
| Batch Size | 128 |
| Learning Rate | $1 \times 10^{-4}$ |
| Early Stopping Patience | 16 |
| Maximum Epochs | 200 |
| Categorical Embedding Dim | 128 |

Table 15: Default Hyperparameters for `AutoInt`

| Hyperparameter | Value |
|---|---|
| Hidden Dimension | 192 |
| Number of Layers | 3 |
| Number of Heads | 8 |
| Attention Dropout | 0.1 |
| Residual Dropout | 0.1 |
| Batch Size | 128 |
| Learning Rate | $1 \times 10^{-4}$ |
| Early Stopping Patience | 16 |
| Maximum Epochs | 200 |

Table 16: Default Hyperparameters for `FT-Transformer`

| Hyperparameter | Value |
|---|---|
| Hidden Dimension | 192 |
| Number of Layers | 3 |
| Number of Heads | 8 |
| Attention Dropout | 0.1 |
| Residual Dropout | 0.0 |
| Batch Size | 128 |
| Learning Rate | $1 \times 10^{-4}$ |
| Early Stopping Patience | 16 |
| Maximum Epochs | 200 |

## F.2 SELF-SUPERVISED METHODS

**SwitchTab** SwitchTab is a framework for tabular data representation learning that utilizes anencoder-decoder architecture to decouple features into mutual (shared across samples) and salient (unique to individual samples) representations. The process begins with feature corruption applied to input data to enhance robustness. The corrupted data is then encoded by an encoder, producing feature vectors that are decoupled into mutual and salient components using two projectors. These components are recombined and reconstructed by a decoder, with both recovered and switched outputs contributing to the computation of a reconstruction loss.

The results presented are drawn from the original work (Wu et al., 2024). Since the code was not available, results for the MNIST dataset were not included.

**BinRecon**  In the approach presented in (Lee et al., 2024), continuous numerical features are discretized into a fixed number of bins, where each bin represents a range of values defined by quantiles of the training dataset. The task is to predict the bin indices instead of reconstructing the raw values, effectively transforming the problem into a regression or classification task depending on whether the bins are treated as ordinal or categorical. This encourages the encoder to learn representations that capture irregularities and nonlinear dependencies characteristic of tabular data. The loss for BinRecon, when treating bins as ordinal values, is defined as:

$$L_{BinRecon} = \frac{1}{N} \sum_{i=1}^{N} \|t_i - f_{BinRecon}(z_i)\|^2,$$

where $t_i$ denotes the bin indices of the input features, $z_i$ represents the encoder outputs, and $f_{BinRecon}$ is the decoder network.

**VIME**  The VIME framework is a self-supervised learning method for tabular data that leverages two pretext tasks: feature estimation, which involves reconstructing the original features, and mask estimation, which focuses on predicting the applied binary mask. A masked sample $\tilde{\mathbf{x}}$ is generated as:

$$\tilde{\mathbf{x}} = \mathbf{m} \odot \bar{\mathbf{x}} + (1 - \mathbf{m}) \odot \mathbf{x},$$

where $\mathbf{m}$ is a binary mask, and $\bar{\mathbf{x}}$ are values sampled from marginal distributions. An encoder $e$ maps $\tilde{\mathbf{x}}$ to $\mathbf{z}$, which is used to predict the mask ($\hat{\mathbf{m}}$) and reconstruct the input ($\hat{\mathbf{x}}$). The framework minimizes:

$$\min_{e, s_m, s_r} \mathbb{E}\left[l_m(\mathbf{m}, \hat{\mathbf{m}}) + \alpha \cdot l_r(\mathbf{x}, \hat{\mathbf{x}})\right],$$

where $l_m$ is binary cross-entropy and $l_r$ is reconstruction loss.

The datasets `AD` and `MNIST` were sourced from the work presented in (Yoon et al., 2020). For the remaining datasets, the publicly available code repositories were utilized, ensuring reproducibility and adherence to standardized evaluation protocols.

The hyperparameters presented in Table 17 were obtained following a hyperparameter tuning process, designed to optimize performance metrics for each specific dataset. This tuning involved Bayesian hyper-parameter search across a range of values for parameters such as `alpha`, `beta`, `mlp_hidden_dim`, and `p_m`, ensuring that the reported results reflect the best possible configurations for the proposed approach.

Table 17: Hyperparameter Settings for VIME experiments

| **Dataset** | alpha | beta | mlp_hidden_dim | p_m |
|---|---|---|---|---|
| JA | 3.1343 | $1.5921 \times 10^0$ | 32 | $1.5536 \times 10^{-1}$ |
| AL | 1.3867 | $1.0233 \times 10^0$ | 256 | $2.4229 \times 10^{-1}$ |
| HE | 4.7568 | $8.1190 \times 10^{-1}$ | 256 | $1.2751 \times 10^{-1}$ |
| HI | 4.7680 | $1.4418 \times 10^0$ | 128 | $2.4642 \times 10^{-1}$ |
| CA | 2.5088 | – | 256 | $1.1916 \times 10^{-1}$ |

**SubTab**  SubTab is a framework for representation learning on tabular data inspired by cropping in image augmentation. It divides tabular data into subsets of features, processed by a shared encoder-decoder architecture.

The results in Table 1 were derived using the `AD` and `MNIST` datasets, sourced from (Ucar et al., 2021). For the other datasets, publicly available code repositories were employed, ensuring reproducibility.

The hyperparameters listed in Table 18 were determined through a hyperparameter tuning process aimed at optimizing performance metrics for each dataset. This process utilized Bayesian hyperparameter search across a predefined range of parameter values.

Table 18: Hyperparameter Settings for SubTab experiments

| Dataset | Dropout Rate | Hidden Layers | Learning Rate | Masking Ratio | N Subsets |
|---------|--------------|---------------|---------------|---------------|-----------|
| HE | 0.1941 | $[1024, 256]$ | $1.0867 \times 10^{-3}$ | 0.2 | 4 |
| JA | 0.1383 | $[1024, 1024, 128]$ | $1.6595 \times 10^{-3}$ | 0.3 | 4 |
| AL | 0.1542 | $[1024, 512]$ | $6.1929 \times 10^{-4}$ | 0.1 | 6 |
| CA | 0.0292 | $[128, 128]$ | $5.8552 \times 10^{-4}$ | 0.1 | 4 |
| HI | 0.1310 | $[512, 512]$ | $1.4141 \times 10^{-3}$ | 0.2 | 4 |

## G    REPRESENTATION SPACE CHARACTERIZATION

**Kullback-Leibler divergence**    The Kullback-Leibler (KL) divergence measures the difference between the probability distributions of random points within the embedding space. Formally, given two probability distributions $P$ and $Q$ over the same variable $x$, the KL-divergence from $Q$ to $P$ is defined as:

$$D_{\text{KL}}(P \parallel Q) = \sum_x P(x) \log \frac{P(x)}{Q(x)}. \tag{8}$$

We consider random points $x_i = \texttt{flatten}(\mathbf{h}_i)$ and $x_j = \texttt{flatten}(\mathbf{h}_j)$ from the embedding space and estimate their distributions $P$ and $Q$, where $\texttt{flatten}$ is an operation that converts the high-dimensional embeddings $\mathbf{h} \in \mathbb{R}^{d \times h}$ into a one-dimensional vector $\mathbf{x} \in \mathbb{R}^{d \cdot h}$. A lower KL-divergence indicates that the embedded space has consistent and similar distributions for different regions. For completeness we also include the euclidean distance between the flattened representations.

**Uniformity**    We rely on the uniformity score (Wang and Isola, 2020) to evaluate the preservation of maximal information within the feature distribution. This score leverages the Gaussian potential kernel $G_t : \mathcal{S}^d \times \mathcal{S}^d \to \mathbb{R}_+$, defined as $G_t(u, v) \triangleq e^{-t\|u-v\|_2^2}$, $t > 0$. The uniformity score is defined as the logarithm of the average pairwise Gaussian potential, formally:

$$\texttt{uniformity} \triangleq -\log \mathop{\mathbb{E}}_{x,y \overset{\text{i.i.d.}}{\sim} p_{\text{data}}} [G_t(u, v)] = -\log \mathop{\mathbb{E}}_{x,y \overset{\text{i.i.d.}}{\sim} p_{\text{data}}} [e^{-t\|u-v\|_2^2}], \ t > 0. \tag{9}$$

This metric is intricately connected to the notion of uniform distribution on the unit hypersphere. This uniformity score allows us to obtain a nuanced measure that captures the degree of information preservation in the feature distribution.

# H  REGULARIZATION TOKEN

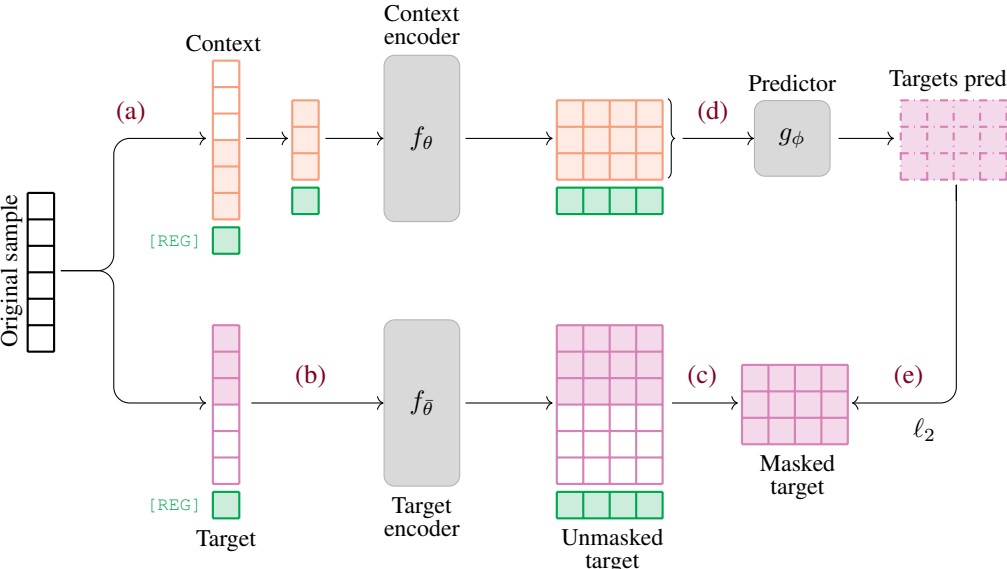

Figure 7: **Regularization Token Pipeline**. In step (a) a sample $\mathbf{x} \in \mathbb{R}^d$ is pre-processed and masked according to a context mask, a [REG] token is appended to the masked representation. In step (b) the whole unmasked sample is pre-processed and fed to the target encoder, including the [REG] token. In step (c) the output of the target encoder is masked according to a target mask (see equation 4). In step (d), the [REG] token is discarded and the remaining features' representations are fed to the predictor to predict the target output. In step (e) we compute the $\ell_2$-distance between the masked target representation and the corresponding prediction.

As discussed in sections 3 and 5, including regularization tokens in the T-JEPA training pipeline is instrumental to escaping representation collapse. While EMA between the weights of the context and target encoders is sufficient for JEPA-based approaches for images (Assran et al., 2023) or videos (Bardes et al., 2024), we show in section 5.2 that without [REG] tokens, the training loss is stuck in a collapsed equilibrium.

We provide in this section a more detailed description on how [REG] are involved in the training pipeline. In particular, we provide in Figure 7 a descriptive example on how the regularization tokens are handled.

**[REG] Token Pipeline**    Recall that $\mathbf{z}_{\mathbf{x}}^{\mathbf{m}} \in \mathbb{R}^{l_{\mathbf{m}} \times h}$ denotes the obtained embedded representation of sample $\mathbf{x}$ with mask $\mathbf{m}$.

- **Context**: Let $\mathbf{m}_c$ denote the context mask. The context representation is obtained by concatenating the masked embedded representation and the embedded [REG] token, $\tilde{\mathbf{z}}_{\mathbf{x}}^{\mathbf{m}_c} = \texttt{concat}(\mathbf{z}_{\mathbf{x}}^{\mathbf{m}_c}, \texttt{[REG]})$, and passing it through the context encoder. One obtains

$$\tilde{h}_{\text{context}}^{\mathbf{m}_c} = f_\theta(\tilde{\mathbf{z}}_{\mathbf{x}}^{\mathbf{m}_c}) \in \mathbb{R}^{(l_{\mathbf{m}_c}+1) \times h},$$

  where $l_{\mathbf{m}_c}$ corresponds to the number of unmasked features in mask $\mathbf{m}_c$, and the supplementary dimension comes from the [REG] token.

- **Target**: The target representation is obtained by concatenating the *unmasked* embedded representation and the embedded [REG] token, $\tilde{\mathbf{z}}_{\mathbf{x}}^{\mathbf{0}_d} = \texttt{concat}(\mathbf{z}_{\mathbf{x}}^{\mathbf{0}_d}, \texttt{[REG]})$ and passing it through the target encoder. One obtains

$$\tilde{h}_{\text{target}} = f_\theta(\tilde{\mathbf{z}}_{\mathbf{x}}^{\mathbf{0}_d}) \in \mathbb{R}^{(d+1) \times h},$$

  Then this representation is masked as detailed in equation 4 using the target mask $\mathbf{m}_t$, to obtain $h_{\text{target}}^{\mathbf{m}_t}$. Note that the [REG] token is always masked/dropped here. See step (c) in Figure 7.

- The `[REG]` token is dropped from the context representation that serves to predict the target representation. One thus obtains $h_{\text{context}}^{\mathbf{m}_c}$,

$$\tilde{h}_{\text{context}}^{\mathbf{m}_c} \xrightarrow{\text{drop } \mathtt{[REG]}} h_{\text{context}}^{\mathbf{m}_c} \in \mathbb{R}^{l_{\mathbf{m}_c} \times h}.$$

  See step (d) in Figure 7.

- **Prediction**: One then uses $h_{\text{context}}^{\mathbf{m}_c}$ to predict $h_{\text{target}}^{\mathbf{m}_t}$,

$$\hat{h}_{\text{target}}^{\mathbf{m}_t} = g_\phi(h_{\text{context}}^{\mathbf{m}_c}, \mathbf{m}_t).$$

Note that, while we experiment using more than one `[REG]` token in section 5.2, the rest of the experiments presented in this work include one `[REG]` token.

