# OpenReview forum: "T-JEPA: Augmentation-Free Self-Supervised Learning for Tabular Data"
_ICLR.cc/2025/Conference — ICLR 2025 Poster_

### Official Review · Reviewer_haUm · 2024-10-30

**Soundness:** 2
**Presentation:** 3
**Contribution:** 2
**Rating:** 5
**Confidence:** 4

**Summary:**

The authors propose a modification of current SSL methods for tabular settings, such as SubTab, and demonstrate the effectiveness of their method on simple baselines; however, they did not conduct a thorough comparison with these baselines.

**Strengths:**

The method is simple and straightforward, addressing an important question.

**Weaknesses:**

- The experiments do not seem to include some relevant baselines, including SubTab, which appears closely related to this methodology, as well as SCARF and others.

- I see that the authors are using the same dataset as the one used in prior work. However, is there a reason for not including datasets used in VIME, SCARF, and SubTab?

- The concept appears very similar to SubTab, at least from a high-level perspective, but the authors did not thoroughly discuss the distinctions. Additionally, the experiments do not demonstrate how the proposed method improves upon SubTab.

- It is unclear how much the method outperforms PTaRL, as the latter shows superior results in several cases. The authors also did not provide sufficient explanation for this, instead placing the results in the appendix.

**Questions:**

- The authors emphasize the term "augmentation-free," but I wonder if previous methods such as SubTab and SCARF are also augmentation-free in the same way defined in this paper.

- There is a workshop paper, *STab: Self-supervised Learning for Tabular Data,* which also claims to be augmentation-free, and it appears they genuinely lack any form of augmentation. How does the current work’s use of “augmentation-free” compare with such approaches? Additionally, a discussion of the differences between these works in this context would be helpful for clarity.

- What are the variances in the reported results?

- It is essential to include comparisons with other SSL methods for tabular data in the main paper.

- Given recent advancements in LLM-based methods and foundation models, it might also be relevant to consider these approaches for this problem.


-----------------------------------------------------------------------
**After Rebuttal:**
Thank you to the authors for conducting new experiments and significantly improving the experimental section compared to the submitted draft. Based on your response, my understanding of the proposed method and its comparison with the baselines was indeed correct. I have increased my score from 3 to 5, reflecting the substantial improvements made during the rebuttal process.

Unfortunately, I am unable to provide further comments on the paper. However, if the authors are able to respond, it would be greatly appreciated:

1- The authors mentioned in their response that "Results from SwitchTab and VIME were directly obtained from their paper." However, there is a mismatch between the reported values in the response section and those in the VIME paper for MNIST. Could you please elaborate on this discrepancy?

2- Could you also provide details about the capacity of each model architecture? The authors stated that in some cases, results were taken directly from the paper, and in others, the available codes were used to run experiments. It is unclear what different architectures were used for each method.

---

### Official Review · Reviewer_NZ9j · 2024-10-30

**Soundness:** 3
**Presentation:** 3
**Contribution:** 3
**Rating:** 8
**Confidence:** 4

**Summary:**

This paper proposes a novel augmentation-free SSL method for tabular data in a model-agnostic way, that relies on a Joint Embedding Predictive Architecture (JEPA) and is akin to mask reconstruction in the latent space.

**Strengths:**

1. The proposed SSL method is model-agnostic and could be coupled with various deep tabular models, which has wide range application.
2. The strategy for masking representations is novel in tabular domain.
3. The experiment is extensive.

**Weaknesses:**

Please refer to the questions below.

**Questions:**

1. What is the meaning of sentence 'We also learn h-dimensional index and feature-type embeddings' in Line 194-195? Please make it more clear.
2. For Regularization token ablation in Fig.4, it seems that without including any regularization token (blue) has the best performance. However, the authors indicate that without including any regularization token, the optimization process does not manage to escape the initial collapse, thereby resulting in a poor performance. Please give more explanations.
3. Except for regularization token ablation on training loss, the authors should further give ablation study on performance, since (1) training loss is not identical to the final evaluation performance and (2) the regularization token is critical for training JEPA-based models on structured data.
4. The sentence 'We construct context and target masks so that there is no overlap between context and target masks but allow overlap within context masks and target masks' in Line 215-216 is confusing. 'no overlap between context and target masks' and 'allow overlap within context masks and target masks' are contradictory.
5. The description about right four subfigures of Fig.3 is not enough.

---

### Official Review · Reviewer_eHM5 · 2024-10-31

**Soundness:** 3
**Presentation:** 2
**Contribution:** 2
**Rating:** 6
**Confidence:** 3

**Summary:**

This paper introduces T-JEPA, a novel self-supervised learning method for tabular data that does not require data augmentations. T-JEPA predicts the latent representation of one subset of features from another subset within the same sample. It operates in the latent space rather than the original data space, avoiding the need for explicit data augmentations. The authors conducts extensive experiments to demonstrate that T-JEPA can serve as a model-agnostic performance boost for tabular learning tasks. The paper also analyzes the reasons behind performance gain through the lens of representation space visualization.

**Strengths:**

1. The introduction of a regularization token is an interesting idea. The authors conduct thorough ablation studies to validate its effectiveness. I appreciate that this technique is not only technically sound but also produces convincing experimental results.
2. The experimental design is comprehensive. The paper evaluates its approach across multiple dimensions: (1) testing on various backbones with and without T-JEPA, (2) comparing backbone+T-JEPA with backbone+PTaRL, (3) benchmarking against other self-supervised models, and (4) contrasting results with supervised decision tree methods. Additionally, the analysis of representation visualization and collapse is thorough.
3. The inclusion of failed framework design attempts in the appendix is particularly valuable. This transparency provides deeper insights and can guide future researchers who are to follow the work.

**Weaknesses:**

1. The paper lacks detailed information about the model architectures used in the main experiments. For example, does ResNet refer to ResNet-18 or ResNet-50? Providing the exact specifications of the hyperparameters can enhance the reproducibility of the experiments.
2. When compared to PTaRL, which is also enhancement for tabular learning, the performance gain is not competitive. Although the idea is novel, the lackluster performance severely limits the contribution of this paper. The authors are encouraged to reimplement PTaRL themselves, rather than simply reusing numbers from previous papers. Though PTaRL has reproducibility challenges due to it not being open-sourced, relying solely on existing published results without additional validation weakens the paper.
3. The description of the methods, specifically how masking is done, is confusing. It would be better to include a figure showcasing the framework of T-JEPA and how downstream task is performed.

**Questions:**

1. Could you provide a figure illustrating the full T-JEPA framework? It would be helpful to show how masking, context encoding, target encoding, and prediction are applied to a sample of tabular data. Simply describing the pipeline may cause confusion for some readers. For example, consider referencing the structure used in Figure 3 of the I-JEPA paper [1].
2. In Figure 4 (a) to (d), the explaination regarding how the representation figure is plotted is needed. Additionally, the paper should include more discussion on the insights derived from these plots. Specifically, what does a more diverse heatmap indicate in your experiments? I found some explanations in Appendix D, but the connection was unclear until I reached that section. At the very least, it would be helpful if the main text refers readers to Appendix D for a more detailed explanation and interpretation of the figure.
3. Is there any correlation between the performance improvements gained from T-JEPA and the number of parameters in the backbone models? Can stronger models with more parameters capture more context information and therefore leading to stronger representation structure and better performance?
4. I suggest the authors include the results comparison with PTaRL to table 1, as the two methods are all enhancements to table learning paradigm.

[1] Self-Supervised Learning from Images with a Joint-Embedding Predictive Architecture. CVPR 2023.

---

### Official Review · Reviewer_xkk6 · 2024-11-01

**Soundness:** 3
**Presentation:** 3
**Contribution:** 2
**Rating:** 6
**Confidence:** 4

**Summary:**

The authors propose a self-supervised representation learning approach for tabular data based on the Joint Embedding Prediction Augmentation (JEPA) framework. Methodologically, the model is based on the work of Assran et al. (2023): inputs are masked, passed to 2 encoders, and a prediction model is trying to reconstruct one of the encoded inputs from the other. Empirically, the authors show that using the features from their model, T-JEPA, increases performance on the downstream tasks. The authors also offer some discussion, highlighting that the identified features are indeed meaningful.

[1] Assran, Mahmoud, Quentin Duval, Ishan Misra, Piotr Bojanowski, Pascal Vincent, Michael Rabbat, Yann LeCun, and Nicolas Ballas. "Self-supervised learning from images with a joint-embedding predictive architecture." In Proceedings of the IEEE/CVF Conference on Computer Vision and Pattern Recognition, pp. 15619-15629. 2023.

**Strengths:**

The authors' idea of applying JEPA for representation learning on tabular data is original. The work is also well-supported by extensive experiments. In general, coming up with a masking strategy to make JEPA work for tabular data constitutes a strong engineering contribution. Tabular data, unlike the other data modalities such as vision, speech, or text, is highly heterogeneous and complex: each feature has its own distribution, some might be categorical while other continuous with varying degrees of non-Gaussianity. Therefore, representation learning on tabular data poses a significant challenge. We have seen a number of works in the recent years in this area. The present work appears to be an important step forward: it finds that JEPA works for tabular data with little modifications, which is promising.

**Weaknesses:**

I am not convinced about the contributions of Sections 5.1 and 5.3. These sections look artificial and drawn-out to me.

## Comments on Section 5.1
The proposed metrics in Section 5.1 have little to do with the optimized objective, Equation 5. Therefore, I have no intuition of what we want these metrics to be: what is their range? what values are good / bad? how fast / slow do we want them to converge? why do we think optimizing our objective promotes this? The authors' arguments such as
> Line 366 "A lower KL-divergence indicates that the embedded space has consistent and similar distributions for different regions."

> Line 377 "This uniformity score allows us to obtain a nuanced measure that captures the degree of information preservation in the feature distribution."

> Line 422 " stable intra-feature variance that indicates consistent information extraction, while the decreasing inter-feature variance shows improved feature specialization"

are imprecise and hand-wavy. If anything, these arguments confuse me more than provide any explanation.

## Comments on Section 5.3
I am not sure I fully grasp what the author do in Section 5.3. As far I as I understand, they want to compare the feature importance produced by XGBoost (and permutation test) to the feature importance produced by their model. However, I don't think the quantity that the authors claim to be feature importance for their method, which is the variance of the hidden features, is a meaningful proxy. The main problem with this metric is the following. Let's say we have feature vector, $\[x_1, x_2\] \in \mathbb{R}^2$, which we encode to get $\[\mathbf{h}_1^T, \mathbf{h}_2^T\]\in\mathbb{R^{2\times h}}$. The authors propose to compute variance of $\mathbf{h}_i$ across hidden dimension and use this quantity to rank $x_i$. However, in general, $\mathbf{h}_i$ does not correspond to $x_i$. Specifically, in the transformer model, when we compute the self-attention, each of $\mathbf{h}_i$ becomes a weighted some of all the other tokens. In the very simple case, consider what happens, with hidden dimension $h=1$ and if the learned encoding is a simple permutation. Clearly, this is not correct.

**Questions:**

1. What is $h$-dimensional index and feature-type embeddings, defined on Line 195?
2. I think Figure 1 does not quite match the description in Section 3. In particular, in Figure 1, I get a sense that there is a single context mask for each sample, while on Line 216 and Line 241, there appears to be multiple context mask for a single sample. In general, how are these masks drawn and how do you ensure that the context / target masks don't overlap?
3. For Equation(5), I think there is a missing superscript $j$ as $\mathbf{m_j} \in M_{context}$.
4. I don't quite follow why we can't do masking for target features before encoding, the same as we do for context. Specifically, this creates data leak, and I suspect that the mode collapse that you demonstrate in Figure 3 is related exactly to this. For a simplified example of why the data leak happens, consider that we have context and target masks, $\mathbf{m}_1$ and $\mathbf{m}_2$ respectively. Further, for simplicity, suppose $\|\mathbf{m}_1\| = |\mathbf{m}_2|$. Then I claim that picking $f\_\theta$ to be the identity map and $f\_{\bar\theta}$ to be the appropriately constructed permutation, we can achieve 0 reconstruction error. For the context features, we have the following transformations: $\mathbf{z}\_{\mathbf{x}} \xrightarrow{\mathbf{m}_1}  \mathbf{z}\_{\mathbf{x}}^{\mathbf{m}\_1} \xrightarrow{f\_\theta}  \mathbf{z}\_{\mathbf{x}}^{\mathbf{m}\_1} $. For the target, features we have: $\mathbf{z}\_{\mathbf{x}} \xrightarrow{f\_{\bar\theta}} \[\mathbf{z}\_{\mathbf{x}, \pi(1)}, \mathbf{z}\_{\mathbf{x}, \pi(2)}, \dots, \mathbf{z}\_{\mathbf{x}, \pi(d)}\] \xrightarrow{\mathbf{m}_2}  \mathbf{z}\_{\mathbf{x}}^{\mathbf{m}\_1}$, where $\pi$ is the constructed permutation. My conjecture is that the models initially learn this, but then are thrown off by the additional regularization. However, overall, I am not sure if the formulation is correct.

---

> ### Comment · Reviewer_xkk6 · 2024-11-15
> **Response after discussion with authors**
>
> Overall, I think the paper presents an effective approach for self-supervised learning based on the JEPA method. However, as it stands, I think the paper could use improvements in a few major areas:
> 1. **Substantiate or refine your claims** throughout the paper, and especially in Sections 5.1 and 5.3. I agree with the authors responses to my concerns on this Sections, but I am afraid incorporating these changes would entail a major rewrite.
> 2. **Deeper investigation** is clearly needed to uncover the causes of the mode collapse. From the reported results, it's clear the training of the model is very sensitive. However, the authors only speculate as to the underlying causes and don't offer any remedies or intuition. I think the addition of further experiments would help make the paper much stronger.
>
> Based on the above, I will keep my score as is.

---

### Meta-Review · Area_Chair_cLLi · 2024-12-12

**Metareview:**

This paper focuses on using self-supervised learning to process tabular data. It proposes a simple but effective method to learning deep representations in an unsupervised manner, without complex augmentation. Experiments are overall comprehensive and well justify the paper's claims.

The strengths of this paper are reflected in several aspects. First, the research problem is interesting and important, but less explored. Second, the technical details of the proposed method are easy to follow. Third, experiments are extensive, especially after rebuttal. The main weakness of this paper is that its technical contribution is somewhat limited. AC checks the paper, review comments, and author feedback, and organizes the discussion with reviewers. Overall, this is a good paper that contributes to the research field of self-supervised learning and would inspire lots of downstream applications. Therefore, it is recommended for acceptance. The author's responses and promises in the rebuttal should be reflected in the final version.

**Additional Comments On Reviewer Discussion:**

During the reviewing process, several concerns were raised by four reviewers, which are mainly reflected in the novelty of this work and its experiments (baselines, models, and other details). The authors provide detailed feedback to address them. After rebuttal, three reviewers are satisfied with the current version and positive about acceptance. One reviewer did not reply.

- Reviewer xkk6 considers that this is a good work with potential and makes a positive impact on the machine learning community.
- Reviewer eHM5 acknowledges the merits of the work and presents a valuable direction for the research community.
- Reviewer NZ9j is positive about this work and thinks the rebuttal addresses the concerns well, and therefore it should be above the acceptance threshold.
- Reviewer haUm points out a some questions and claims that lots of them are addressed. Besides, after checking the questions and final feedback by authors, the rebuttal is satisfactory.

In summary, although the technical novelty of the proposed method is somewhat limited and needs to be improved, in other respects, this paper is considered to meet the acceptance line. Therefore, AC makes an acceptance recommendation finally.

---

### Decision · Program_Chairs · 2025-01-22

Accept (Poster)